# Identification of a fungal antibacterial endopeptidase that cleaves peptidoglycan

Silke Machata[1], Ute Bertsche [ID][2], Franziska Hoffmann[3], Zaher M Fattal[1], Franziska Kage [ID][1], Michal Flak[4], Alexander N J Iliou[5], Falk Hillmann [ID][5,6], Ferdinand von Eggeling [ID][3], Hortense Slevogt[7,8], Axel A Brakhage [ID][4,9] & Ilse D Jacobsen [ID][1,9][✉]

## Abstract

*Aspergillus fumigatus* is a saprophytic fungus dwelling in soil and on decaying plant material, but also an opportunistic pathogen in immunocompromised patients. In its environmental niche, *A. fumigatus* faces competition from other microorganisms including bacteria. Here, we describe the discovery of the first secreted antibacterial protein in *A. fumigatus*. We identify a secreted fungal endopeptidase, designated CwhA, that cleaves peptidoglycan of Gram-positive bacteria at specific residues within the peptidoglycan stem peptide. Cleavage leads to bacterial lysis and the release of peptidoglycan cleavage products. Expression of *cwhA* is induced by the presence of bacteria. Furthermore, CwhA is highly abundant in murine lungs during invasive pulmonary aspergillosis and peptidoglycan cleavage products generated by CwhA stimulate cytokine production of human immune cells in vitro. Although CwhA does not affect human cells directly, this novel player in fungal-bacterial interactions could affect *A. fumigatus* infections by inhibiting Gram-positive bacteria in its vicinity, and possibly modulate the immune system.

**Keywords** *Aspergillus fumigatus*; Fungal–Bacterial Interactions; *Staphylococcus aureus*; Cell Wall
**Subject Categories** Immunology; Microbiology, Virology & Host Pathogen Interaction; Signal Transduction

## Introduction

Saprophytic filamentous fungi such as *Aspergillus* (*A.*) *fumigatus* share their environmental niche in soil and on decaying plant material with a plethora of other microorganisms. In order to survive in this highly competitive environment, filamentous fungi produce a variety of secondary metabolites, including antibiotics, whose production is often induced by the presence of other microorganisms (Adnani et al, 2017; Fischer et al, 2018; Netzker et al, 2015; Netzker et al, 2018; Schroeckh et al, 2009). Some of these molecules also play a role in infections with *A. fumigatus* that occur in immunocompromised patients and patients with underlying lung disease such as cystic fibrosis (CF). One example is gliotoxin, a mycotoxin with amoebicidal properties that negatively affects immune cells and thereby contributes to virulence (Hillmann et al, 2015a; Kosmidis and Denning, 2015; Scharf et al, 2016). Another example is the *A. fumigatus* pigment DHN-melanin that affects phagocytosis and killing of the fungus by both environmental amoeba and mammalian macrophages (Akoumianaki et al, 2016; Ferling et al, 2020).

Polymicrobial communities are not only found in the environment but also in human and animal hosts. Probably best studied is the complex network of the gastrointestinal microbiota and mycobiota, which was shown to have a strong impact on human health (Krüger et al, 2019). In contrast, microorganisms colonizing the lower respiratory tract have been largely neglected in the past, and the healthy lung was long considered to be a sterile environment (Barcik et al, 2020; Levy et al, 2017; Man et al, 2017). This view has changed due to the advances in culture-independent techniques for microbiological analysis, recognizing the complexity of respiratory microbial communities (Chang et al, 2020). Characterized by comparatively low microbial density ($10^3$/g of lung tissue) the lung microbiome composition was shown to change drastically in case of chronic lung diseases such as CF or asthma, leading to dysbiosis and pulmonary infections (Chotirmall and McElvaney, 2014; Dickson et al, 2015; Mathieu et al, 2018; Mitchell and Glanville, 2018). In CF patients, polymicrobial infections often occur that vary in composition and diversity depending on the patient´s age. The majority of adult CF patients' lungs are colonized by *Staphylococcus* (*S.*) *aureus*, the *Burkholderia cepacia* complex, and *Pseudomonas* (*P.*) *aeruginosa* (Blanchard and Waters, 2019). Among fungi, *A. fumigatus* and *Candida albicans* are the most prevalent colonizers in the airways of CF patients (Chotirmall and McElvaney, 2014). In this setting, interactions between *A. fumigatus* and colonizing bacteria are likely (Poore et al, 2021) and have been studied in some detail for *P.*

[1]Microbial Immunology, Leibniz Institute for Natural Product Research and Infection Biology – Hans Knoell Institute, Jena, Germany. [2]University of Hohenheim, Core Facility - Module 1 Mass Spectrometry Unit, Stuttgart, Germany. [3]Department of Otorhinolaryngology, Jena University Hospital, Jena, Germany. [4]Molecular and Applied Microbiology, Leibniz Institute for Natural Product Research and Infection Biology – Hans Knoell Institute, Jena, Germany. [5]Evolution of Microbial Interactions, Leibniz Institute for Natural Product Research and Infection Biology – Hans Knoell Institute, Jena, Germany. [6]Wismar University of Applied Sciences, Wismar, Germany. [7]Department of Respiratory Medicine and Infectious Diseases, Hannover Medical School, German Center for Lung Research (DZL), BREATH, Hannover, Germany. [8]Respiratory Infection Dynamics Group, Helmholtz Centre for Infection Research, Braunschweig, Germany. [9]Institute for Microbiology, Friedrich-Schiller-University Jena, Jena, Germany. [✉]E-mail: ilse.jacobsen@leibniz-hki.de

*aeruginosa*. This Gram-negative pathogen can suppress the growth of *A. fumigatus* by various mechanisms, including the production of phenazines, secondary metabolites that also act as virulence factors, and siderophore-mediated iron depletion, limiting fungal access to essential nutrients. (Briard et al, 2015; Keown et al, 2020). In addition, *Klebsiella pneumoniae* was reported to inhibit spore germination and hyphal development of *Aspergillus* spp. in vitro (Nogueira et al, 2019). Conversely, a recent microbiome analysis suggests that *A. fumigatus* colonization shapes the lung microbiome (Mirhakkak et al, 2023). This likely involves *A. fumigatus* secondary metabolites, several of which have been shown to be antibacterial and whose production is affected by interactions with bacteria (Boysen et al, 2021; Krespach et al, 2023; Margalit et al, 2022; Netzker et al, 2018; Stroe et al, 2020). One example for a secondary metabolite with antibacterial activity is gliotoxin, which inhibits *P. aeruginosa* in coculture experiments (Reece et al, 2018). Another is fumigermin that inhibits germination of bacterial spores (Stroe et al, 2020). Moreover, defensin-like peptides called afusins are produced during *A. fumigatus* conidiation and mediate protection of conidia against bacteria (Dümig et al, 2021). However, no reports of proteins produced by *A. fumigatus* mycelia that could play a role in multispecies interaction by suppressing bacterial growth exist thus far.

Most eubacteria are surrounded by the so-called peptidoglycan (PG) sacculus that is responsible for cell shape and counteracts the internal pressure (Weidel and Pelzer, 1964). It is a macromolecule composed of β-1,4 linked *N*- acetylglucosamine (NAG) and *N*-acetylmuramic acid (NAM) which form glycan strands of variable length. Attached to the lactyl group of MurNAc is a stem peptide of five amino acids with a quite conserved sequence: L-alanine – D-iso-glutamate – L-lysine – D- alanine – D-alanine for Gram-positive organisms and meso-Diaminopimelic acid (mDpm) on position three of Gram-negatives and also of *Bacillus* species (Schleifer and Kandler, 1972). There are several possible modifications known. For example, glycan strands can be O-acetylated, O-deacetylated or N-deacetylated. These modifications often result in lysozyme resistance (Bera et al, 2006; Bera et al, 2005; Crisóstomo et al, 2006). Cross-linking of the glycan strands occurs between two adjacent stem peptides by formation of a peptide bond between D-alanine on position four and the amino acid on position three (L-Lys or mDpm), catalyzed by the penicillin-binding proteins and release of the fifth amino acid D-alanine (Goffin and Ghuysen, 1998; Warth and Strominger, 1971). Cross-linking can be either direct or indirect by an interpeptide bridge, i.e., the pentaglycine bridge in *S. aureus* (Schleifer and Kandler, 1972). For analysis by HPLC, PG is isolated and digested by the muraminidase mutanolysin into muropeptides, thereby cleaving the glycosidic bond (Atrih et al, 1999; Atrih et al, 1996; de Jonge et al, 1992; Glauner, 1988). The resulting disaccharide units can still be cross-linked, thereby forming dimers, trimers and multimers whose retention times increase (Kühner et al, 2014). Digestion of the PG macromolecule into smaller units also occurs in nature and is a defense mechanism of many organisms. Resulting small muropeptides that miss the NAG-part are immunogenic: Muramyl-dipeptide is sensed by NOD2 while the mDpm containing muramyl-tripeptide is recognized by NOD1 (Chamaillard et al, 2003; Girardin et al, 2003a; Girardin et al, 2003b).

In our study presented here, we describe an antibacterial fungal protein, designated CwhA, which is highly abundant in the lungs of mice infected with *A. fumigatus* (Machata et al, 2020). We demonstrate that *cwhA* gene expression is induced by the presence of bacteria and during late stages of infection of alveolar epithelial cells. The protein acts as an endopeptidase degrading PG of Gram-positive bacteria, leading to bacterial lysis and the release of a PG cleavage product that stimulates cytokine production of human immune cells. This novel player in fungal–bacterial interactions could affect *A. fumigatus* infections by inhibiting Gram-positive bacteria in its vicinity, and modulating the immune system.

# Results

## The *A. fumigatus* protein CwhA is produced during infection and contains functional domains of bacterial cell wall hydrolases

We previously performed proteomic analysis of bronchoalveolar lavage samples from mice with experimental invasive pulmonary aspergillosis (Machata et al, 2020). Two of the most highly abundant proteins (B0YAY0, encoded by AFUB_086210 aka AfuA 8g00360, and B0Y269, encoded by AFUB_061470) were annotated as putative NlpC/P60-like cell-wall peptidases but not yet characterized. The respective genes were likewise shown to be induced in a mouse infection model by transcriptome analysis (Kale et al, 2017; McDonagh et al, 2008). The gene encoding B0YAY0 is located on chromosome 8 bordering the fumagillin gene cluster and was found to be transcriptionally upregulated in LaeA (1.42-fold) and SltA (37-fold) deletion mutants, but not by FapR (Liu et al, 2021; Perrin et al, 2007; Wiemann et al, 2013) in vitro. While the corresponding protein was not identified in most previous proteome studies of *A. fumigatus* grown in vitro, it was upregulated in the secretome of a PrtT mutant, but not a PrtT/XprG double mutant (Shemesh et al, 2017).

According to in silico analysis the protein contains a signal peptide responsible for protein secretion (1-35 aa), a bacterial SH3 domain (Pfam PF08239, 21-97 aa), and a protein domain of the NlpC/p60 family (Pfam PF00877, 141–247 aa) (Fig. 1A). Proteins of this superfamily have been mainly described in bacteria and exhibit functional diversity (Anantharaman and Aravind, 2003). The NlpC/p60 domain carries a catalytic triad that consists of a cysteine, a histidine and a polar residue and is often found in cell wall hydrolases of bacteria (Machata et al, 2005; Vermassen et al, 2019; Xu et al, 2015). NlpC/p60 cell wall hydrolases can act as lytic enzymes that permeabilize the bacterial peptidoglycan (PG) layer and potentially lead to lysis. Therefore, we speculated that due to its structural similarity with bacterial cell wall hydrolases, B0YAY0 might be able to degrade PG, and designated the protein CwhA (cell wall hydrolase A) in this study.

Next, we confirmed the production of B0YAY0 during invasive pulmonary aspergillosis (IPA) in mice by MALDI-imaging mass spectrometry (MSI) of lung slices: Six m/z-values were detected by MSI, which correlated to B0YAY0 peptide fragments predicted by in silico digestion (Appendix Table S1). These m/z-values were also detected in control spots containing the recombinant protein. As an example, the distribution of one peptide with an m/z mass value of 1174.334 corresponding to the peptide (K)KEYNTLECR(G) $[M + NH4]^+$ is shown in Fig. 1. The protein was heterogeneously distributed in areas of hyphal growth, implying secretion of the protein from the fungal mycelia (Fig. 1B).

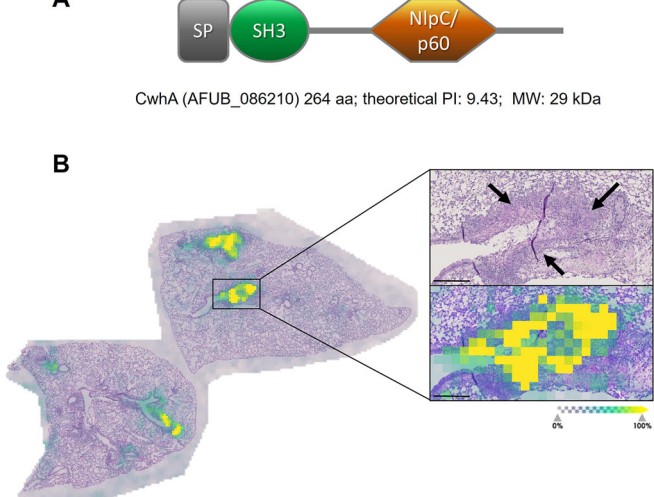

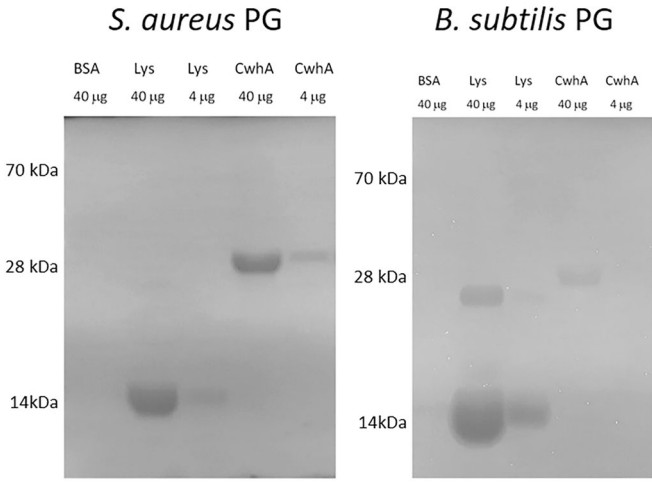

**Figure 1.  Structure and detection of the secreted *A. fumigatus* protein BOYAYO (CwhA) in infected murine lungs by MALDI-imaging.**

(**A**) Protein domains as predicted by Pfam; *A. fumigatus* CwhA contains a signal peptide (SP), SH3 bacterial domain (PF08239) and NlpC/p60 family domain (PF00877). (**B**) Lung histology of leukopenic mice infected with *A. fumigatus*. Fungal mycelia are visualized with periodic acid-Schiff staining and marked by black arrows in the upper enlarged image. The fungal protein was detected as a mass of m/z 1174.704 corresponding to the peptide (K)KEYNTLECR(G) with a mass of 1156.311 [M + NH4]. The lower enlarged image displays the relative abundance of the fungal protein as colored areas. Scale bar picture left: 1 mm; scale bar inserts: 200 μm. $n = 5$ lungs, two sections of each lung were analysed; shown is a representative image. Source data are available online for this figure.

**Figure 2.  Visualization of cell wall hydrolytic activity of CwhA.**

Hydrolytic activity of CwhA against peptidoglycan (PG) from *S. aureus* and *B. subtilis* was visualized by zymography. CwhA, BSA (negative control) and lysozyme (Lys; positive control) were applied in the indicated concentrations to zymogram gels containing bacterial cell wall substrate. Lytic activity leads to clear bands in blue-stained gels. The pictures are inverted to mark cleavage products as dark bands. Representative images of $n = 3$ technical replicates using two different protein preparations are shown. Source data are available online for this figure.

## A. fumigatus CwhA cleaves bacterial PG

To determine if CwhA has hydrolytic activity against bacterial PG, we produced a recombinant protein by replacing the signal peptide sequence with an N-terminal His-tag. This protein fusion was produced in *Escherichia coli* and purified by affinity chromatography. The recombinant protein was tested for its activity using zymogram gels containing PG from *S. aureus* and *Bacillus* (*B.*) *subtilis* as substrates. Cleavage of PG by the activity of the protein resulted in a band with an expected molecular mass of 29 kDa. CwhA was highly active against cell wall material from *S. aureus* and moderately active against *B. subtilis* (Fig. 2). In comparison, the control enzyme lysozyme showed strong activity against *B. subtilis* and was less active against *S. aureus* PG. Based on these findings, we conclude that CwhA acts as a cell wall hydrolase against bacterial PG with particular substrate preference.

## CwhA promotes the lysis of Gram-positive bacteria

To assess the impact of the protein on live bacteria, we treated bacterial cultures of *S. aureus* with recombinant CwhA at various concentrations (16 to 200 μg/ml) and tested bacterial growth by measuring the optical density (OD). While bacterial growth was unaffected when CwhA was added to LB media (Fig. EV1A), we observed a clear decrease of the OD when bacteria were suspended in Tris buffer (50 mM Tris pH 8.0) and treated with CwhA (Figs. 3A and EV1B). Due to the hypoosmotic conditions in the buffer,

damaging effects on the bacterial cell wall are likely to have a stronger impact on the integrity of the bacterial cell compared to complex growth media. Already 10 min upon exposure to CwhA, the OD was considerably reduced compared to the PBS control, showing progressively stronger effects with increasing protein concentrations. Within two hours of treatment, the OD dropped between 30% and 50% in a dose-dependent manner, while a decrease of only 10% could be observed in untreated bacterial cells in the PBS and BSA controls (Figs. 3A and EV2A). In addition to the *S. aureus* strain SA113, we tested methicillin-resistant *S. aureus* strains (MRSA) for their susceptibility to CwhA. CwhA reduced the OD of these MRSA strains to a similar extent as observed for the SA113 reference strain (Fig. 3B; Appendix Table S2). Other Gram-positive bacteria, such as *Enterococcus faecalis* and *Streptococcus pneumoniae*, were also susceptible to CwhA-mediated lysis, while the Gram-negative bacteria *Pseudomonas aeruginosa* and *Klebsiella pneumoniae* were resistant (Fig. EV2B). In contrast to the other tested Gram-positive bacteria, we observed no OD decrease of *B. subtilis* upon treatment of the culture with CwhA (Fig. EV2B). This observation coincides with the zymogram data showing only weak activity of CwhA against *B. subtilis* cell wall substrate and suggests a substrate preference of CwhA for peptidoglycan from *S. aureus* over *B. subtilis* or Gram-negative bacteria.

To test whether CwhA activity leads to the killing of *S. aureus*, bacterial growth was assessed on LB agar plates after treatment of bacteria with CwhA for different time periods (Fig. 3C). 15 min after the addition of CwhA, only 60% of the initial CFU were recovered, while both PBS and lysozyme only led to a minor reduction (~10%). Treatment with CwhA led to further reduction of CFU over time, and after 120 min, nearly all staphylococci were killed. In contrast, after the initial drop, CFU in the PBS control remained stable, and only 50% CFU

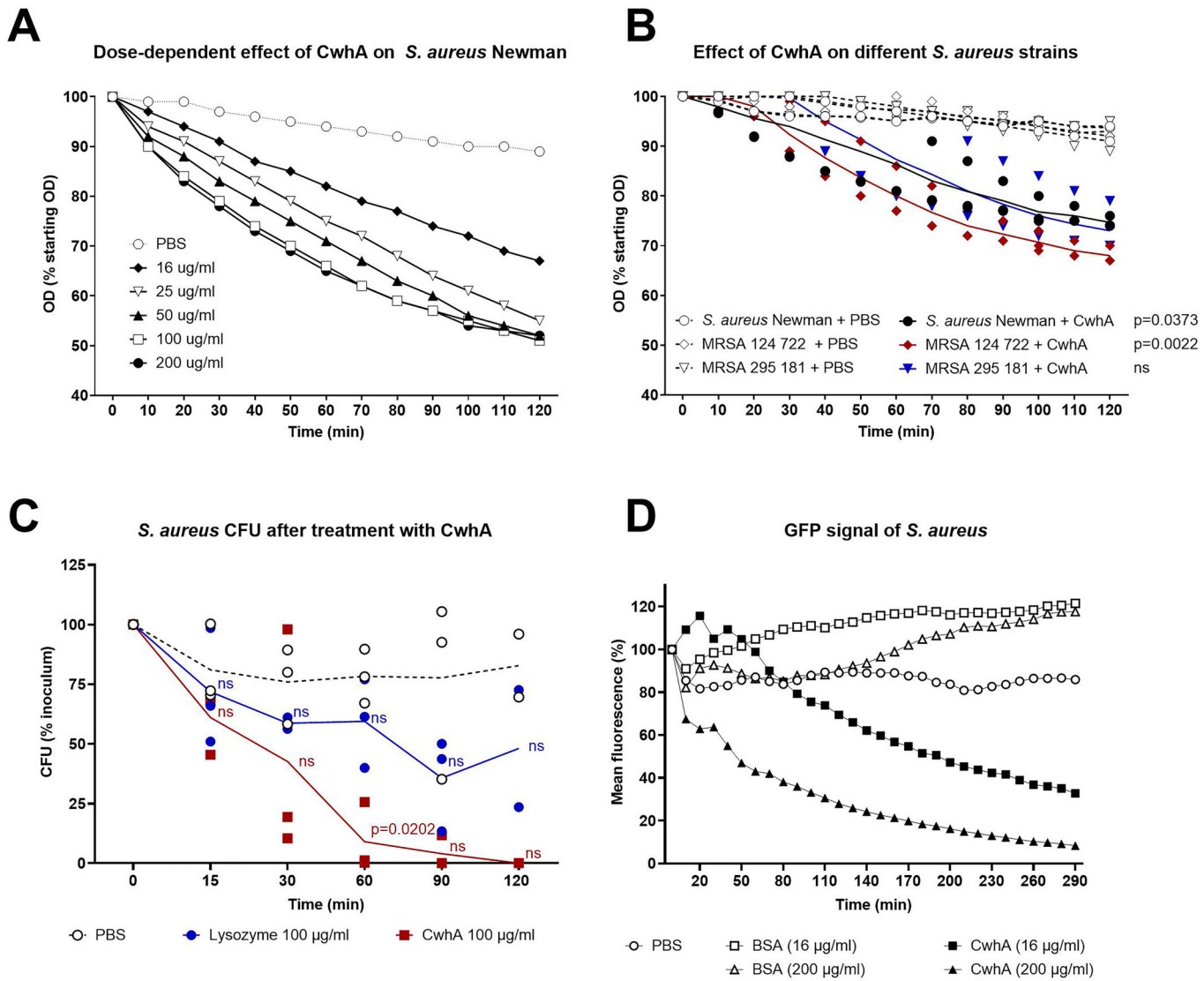

Figure 3.  Effect of recombinant CwhA on *S. aureus*.

(A) The optical density (600 nm) after addition of recombinant CwhA (16–200 µg/ml) to *S. aureus* strain Newman suspensions in 50 mM Tris buffer (pH 8) was measured over time; one of $n = 2$ biological replicates, see also Fig. EV1B. (B) Decrease of optical density of *S. aureus* strain Newman and two MRSA strains after addition of 100 µg/ml CwhA; $n = 3$ biological replicates. (C) The colony-forming units (CFU) of *S. aureus* were determined at the indicated time points after addition of PBS, 100 µg/ml CwhA or 100 µg/ml lysozyme. $n = 3$ biological replicates except 120 min $n = 2$. (D) GFP signal of a GFP-expressing *S. aureus* strain after treatment with CwhA. Bovine serum albumin (BSA) and PBS were used as controls. One of $n = 2$ biological replicates, see also Fig. EV1C. Time-lapse videos of *S. aureus*::Gfp with and without CwhA treatment are supplied as Movies EV1 and EV2. Data information: In (B, C) data were shown as individual data points and the mean as connecting line. Data were statistically analysed by two-way RM ANOVA comparing the control with treatment for each strain (B) or mixed-effects analysis and Šídák's multiple comparisons test comparing the control with each treatment for each timepoint (C). Test results are indicated in the graph, ns not significant. Source data are available online for this figure.

reduction were achieved with lysozyme. To assess if bacterial killing was mediated by cell wall lysis, the effect of CwhA on *S. aureus* was analysed using an *S. aureus* strain expressing green fluorescent protein (GFP) in the cytoplasm. Lysis of bacterial cells should lead to leakage of cytoplasmic contents and thereby loss of fluorescence, which can be visualized by automated live cell imaging microscopy. While the fluorescence of *S. aureus* cells treated with PBS or BSA remained stable, the GFP signal rapidly decreased by 50% within 1 h in the CwhA-treated cells (Figs. 3D and EV2C; Movie EV1). This indicates that CwhA indeed has a bacteriolytic effect on staphylococci, leading to the rupture of the bacteria and release of cytoplasmic content.

## CwhA produced by *A. fumigatus* lyses *S. aureus* in coculture

To determine if the native CwhA protein produced by *A. fumigatus* shows the same lytic activity against *S. aureus* as demonstrated for the recombinant protein, we generated a mutant strain deficient in CwhA (ΔcwhA), as well as a mutant strain that overproduces the protein, using the constitutively active promoter of the *A. fumigatus* gpdA gene (PgpdA::cwhA). While no expression of cwhA was detectable in the deletion mutant, *A. fumigatus* PgpdA::cwhA showed highly increased cwhA gene expression compared to the

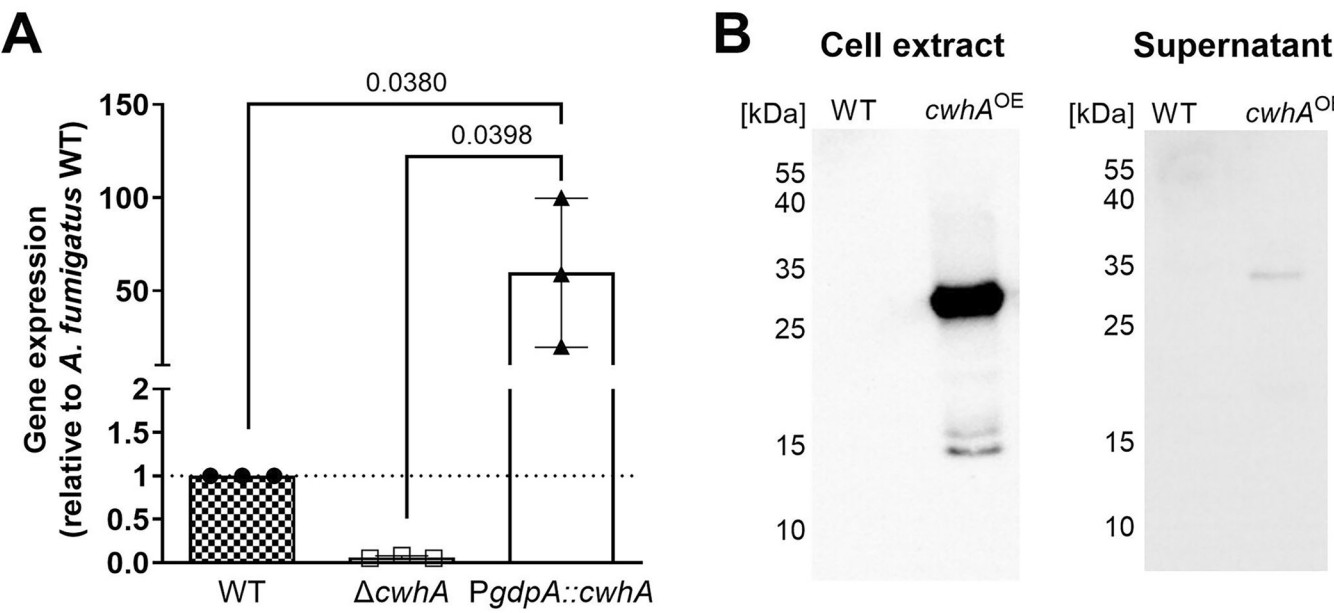

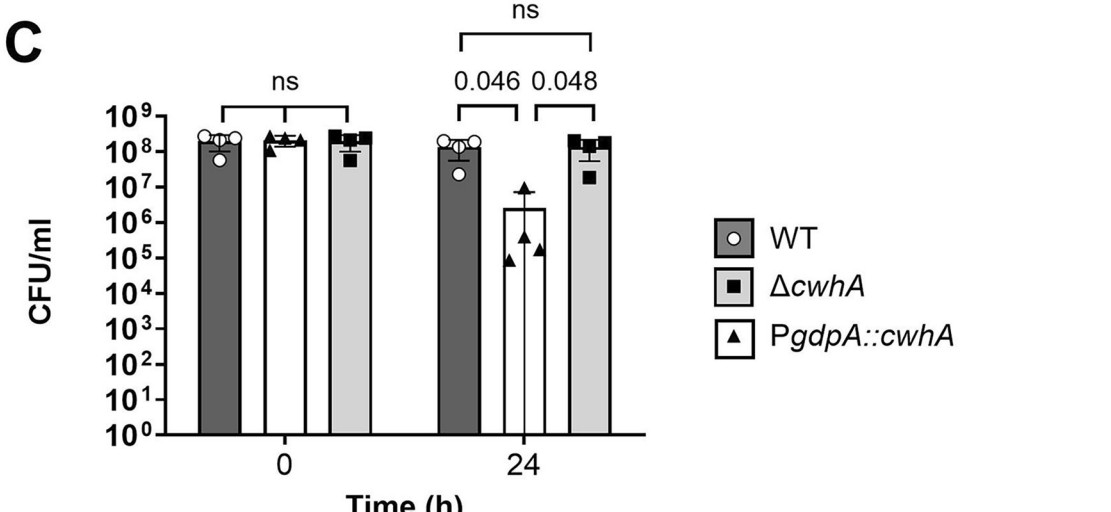

**Figure 4. Overexpression of *cwhA* in *A. fumigatus* increases bacterial lysis.**

(A) Expression of *cwhA* determined by quantitative RT-PCR of *Aspergillus fumigatus*. Relative expression of *cwhA* was determined by the $2^{-\Delta\Delta Ct}$ method with *cox5* and *act1* as housekeeping genes; expression in *A. fumigatus* CEA17 Δ*akuB*$^{KU80}$ (WT) was set to 1 ($n = 3$ biological replicates). (B) CwhA protein expression was determined in whole cell extracts after 1 day of growth and culture supernatants after 3 days of growth of the *A. fumigatus* CEA17 Δ*akuB*$^{KU80}$ strain (WT) and its isogenic overexpressing mutant P*gpdA*::*cwhA* using a monoclonal antibody raised against recombinant CwhA protein. Single experiment. (C) Survival of *S. aureus* after 24 h co-incubation in Tris Buffer with *A. fumigatus* CEA17 Δ*akuB*$^{KU80}$. $n = 4$ biological replicates. Data information: Data in (A, C) are shown as a scatter plot with bars (mean) and error bars (SD). Data were analysed by one-way ANOVA and (A) Šídák's multiple comparisons test comparing the overexpressing strain P*gpdA*::*cwhA* to the parental strain CEA17 Δ*akuB*$^{KU80}$ strain (WT) and the deletion mutant Δ*cwhA* or (C) Tukey's multiple comparisons test. *P* values <0.05 are indicated in the graph, ns not significant. Source data are available online for this figure.

wildtype (Fig. 4A). Western blot analyses using a monoclonal antibody directed against CwhA confirmed protein overexpression in *A. fumigatus* P*gpdA*::*cwhA* (Fig. 4B), which was especially pronounced in cellular extracts. Secretion of the protein, however, was relatively weak during growth in *Aspergillus* minimal medium (AMM) (Fig. 4B). We also confirmed our previous literature-based assumption that the

protein is not produced at conventional laboratory in vitro culture conditions, as no protein band was detectable in protein extracts of the *A. fumigatus* wildtype (Fig. 4B). To determine the functionality of CwhA produced by *A. fumigatus*, the three strains were tested for their capacity to induce bacterial lysis in *S. aureus*. We observed a strong decrease in bacterial numbers after 24 h co-incubation of bacteria with

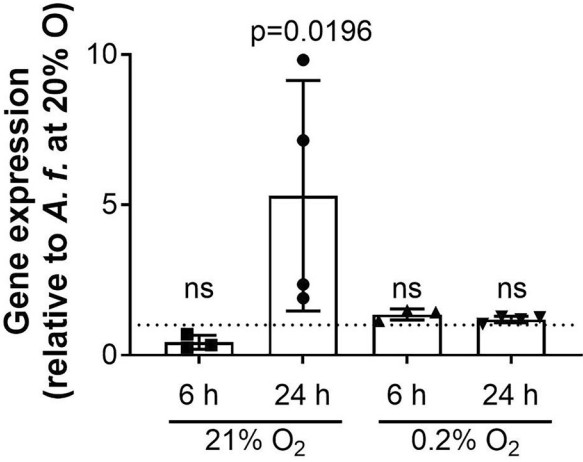

**A**

## *cwhA* expression during infection

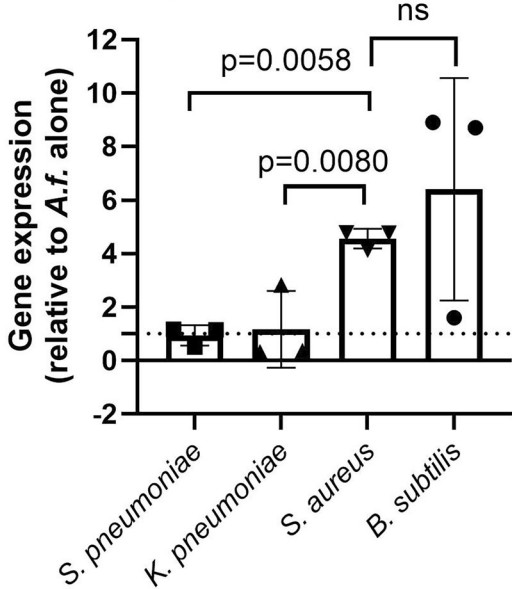

**C**

## *A. fumigatus* - bacteria co-culture

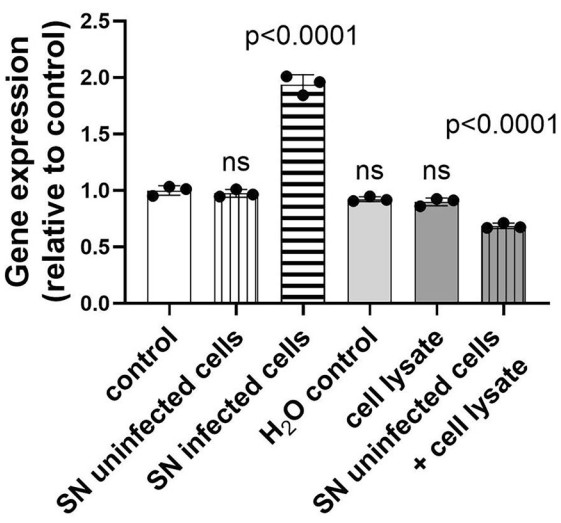

**B**

## *cwhA* expression with supernatants and host cell lysate

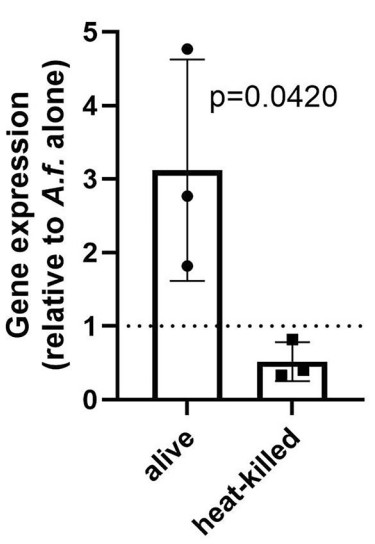

**D**

## Bacterial viability

fungal mycelia of the P*gpdA*::*cwhA* strain (Fig. 4C). In contrast, there were no differences in the bacterial counts after co-cultivation with fungal mycelia of the wildtype and the Δ*cwhA* deletion strain. In conclusion, CwhA produced as a native protein by *A. fumigatus* wild type shows antibacterial features similar to the recombinant protein, but the gene is only weakly expressed in axenic laboratory cultures.

## CwhA expression in *A. fumigatus* is induced by the presence of Gram-positive bacteria

CwhA was identified as one of the most abundant fungal proteins in BAL samples in mice with invasive aspergillosis (Machata et al, 2020). Furthermore, a fivefold upregulation of the expression of Afu8g00360, the gene encoding CwhA in *A. fumigatus* Af293, was

**Figure 5. Impact of oxygen, host cells, and bacteria on *cwhA* expression.**

*cwhA* expression was determined by qRT-PCR using RNA isolated from *A. fumigatus* mycelia. (A) A549 cells were infected with conidia for 9 h at 21% oxygen before washing with fresh media (24 h timepoint only) and placing the plates at either 21 or 0.2% oxygen for the indicated duration. Data were normalized to *A. fumigatus* mycelia cultured at 21% oxygen in the absence of host cells (control). 6 h: $n = 3$, 24 h: $n = 4$ biological replicates. (B) Mycelia were exposed for 4 h to fresh cell culture media (control), supernatants of uninfected A549 cells (SN uninfected cells), supernatants of A549 cells infected with *A. fumigatus* (SN infected cells), cell culture media supplemented with lysed A549 cells (cell lysate) or $H_2O$ ($H_2O$ control), or supernatants of uninfected A549 cells supplemented with cell lysate (SN uninfected cells + cell lysate). $n = 3$ biological replicates. (C) *A. fumigatus* was co-cultured in DMEM with *B. subtilis*, *S. pneumoniae*, *K. pneumoniae*, and *S. aureus*, respectively, for 4 h; $n = 3$ biological replicates. (D) *A. fumigatus* was cultured in DMEM for 4 h in the presence of living or heat-killed *S. aureus* (Newman); $n = 3$ biological replicates. Data information: Data in (A–D) were presented as a scatter plot with mean (bar) ± SD (error bars). Data were statistically analysed by ordinary one-way ANOVA and Dunnett's multiple comparisons test to compare all groups with the control (A, B) or two-tailed unpaired *t*-test (C, D). *P* values <0.05. are indicated in the graph, ns not significant. Source data are available online for this figure.

shown in a previous study by in vivo transcriptome analysis of BAL samples from infected mice (McDonagh et al, 2008). To identify possible factors that lead to the induction of CwhA in vivo, we analysed its gene expression under two conditions that mimic environmental factors in the lung during IPA. First, we tested if oxygen availability has an impact on *cwhA* expression, as hypoxic conditions occur in infected lungs (Grahl et al, 2011; Hillmann et al, 2015b). Our qRT-PCR data revealed no significant difference in the transcript levels of CwhA between cultures exposed to normoxia (21% $O_2$) and hypoxia (0.2% $O_2$; expression 1.067 ± 0.48 compared to normoxia). Secondly, expression in the presence of host cells was analysed. Infection of the human alveolar epithelial cell line A549 for 6 h did not lead to an increase of *cwhA* expression (Fig. 5A). However, a fivefold increase in *cwhA* expression was observed when the infection period was extended to 24 h, at which host cells were partially disrupted by invading fungal mycelia (Fig. EV3A). These data show that reduced oxygen availability has no impact on the *cwhA* gene induction, and that the sole presence of host cells is insufficient to promote gene expression. Because increased expression of *cwhA* coincided with host cell damage, we tested the effect of supernatants collected from non-infected cells, infected cells, and lysed host cells on *cwhA* gene expression. Neither supernatants from uninfected host cells, nor host cell lysates, or a combination thereof led to an increase in *cwhA* expression. However, fungi exposed to the supernatant of host cells infected with *A. fumigatus* showed a 1.5-fold increase (Fig. 5B).

As we found CwhA to be effective against some Gram-positive bacteria, we next assessed whether fungal proximity to bacteria promotes expression of *cwhA* (AFUB_086210). Co-incubation of *A. fumigatus* with either *S. aureus* and *B. subtilis* led to a clear increase in *cwhA* expression (4- and 6-fold, respectively), while the expression remained unaffected by co-incubation with *S. pneumoniae* or *K. pneumoniae* (Fig. 5C). The induction was dependent on bacterial viability as determined with heat-inactivated *S. aureus* (Fig. 5D). Thus, the mere presence of bacterial surface structures does not seem to be sufficient for inducing *cwhA* expression. We also tested if soluble compounds released by bacteria, or changes in the culture environment due to bacterial metabolism, stimulate the expression of *cwhA*. Therefore, fungal mycelia were incubated in sterile-filtered supernatants of bacterial and fungal–bacterial cocultures. Neither type of supernatant led to increased *cwhA* expression (Fig. EV3B), indicating that soluble factors and nutritional starvation alone are not the sole inducing factors, and that a combination of bacterial activity and physical interaction might be required.

## Recombinant CwhA causes no overt damage to amoeba, host cells, and fungal cells

We next assessed the role of CwhA in interaction with organisms other than bacteria that can occur in close proximity to *A. fumigatus* in nature and might be affected by the secreted protein. Amoebae are commonly found in damp soil where they co-localize with *A. fumigatus* (Hillmann et al, 2015a). We therefore tested the susceptibility of the amoeba *Dictyostelium discoideum* to CwhA. At early time points, the recombinant protein had a minor growth-promoting effect, whereas after 26 h, slightly fewer amoeba were observed in the presence of CwhA compared to the control (Fig. 6A). However, amoeba numbers increased approximately tenfold from 1.5 to 26 h both in the presence and absence of CwhA, arguing against an overt cytotoxic effect of CwhA. Since *A. fumigatus* CwhA was produced in detectable amounts in lung tissue during murine IPA, we next determined if CwhA contributes to host cell damage. CwhA concentrations ranging from 0.4 to 25 μg/ml did not cause detectable necrotic cell damage of murine alveolar macrophages (MH-S) and human pulmonary epithelial cells (A549) within 12 h of exposure (Fig. 6B). Last, we also analysed if CwhA has an inhibitory effect on *A. fumigatus* grown as a planktonic culture or as a biofilm. Addition of recombinant CwhA to *A. fumigatus* in AMM had a stimulatory effect on growth if compared to the PBS control, comparable to the control protein bovine serum albumin (BSA) (Fig. 6C,D). This likely indicates that CwhA is utilized as an additional nutrient source in these settings. In summary, recombinant CwhA exhibits lytic activity against some Gram-positive bacteria, but not against the Gram-negative bacteria tested, amoebae, host cells, and *A. fumigatus* itself.

## CwhA acts as an endopeptidase on the peptidoglycan of Gram-positive bacteria, containing ʟ-lysine in the peptidoglycan stem peptide

The observation that recombinant CwhA was only active against certain bacteria suggested substrate specificity. Although both *S. aureus* and *B. subtilis* are Gram-positive, they differ in their PG composition. The most important difference is in position three of the stem peptide, where *S. aureus* harbors ʟ-lysine, which is typical for Gram-positives. *B. subtilis* contains meso-diaminopimelic acid (mDpm) instead, normally found in the PG of Gram-negatives (Atrih et al, 1999; Do et al, 2020; Kim et al, 2015; Vollmer et al, 2008). To determine the exact hydrolytic specificity of CwhA, PG isolated from *S. aureus* and *B. subtilis* was treated with the muraminidase mutanolysin alone or with a combination of

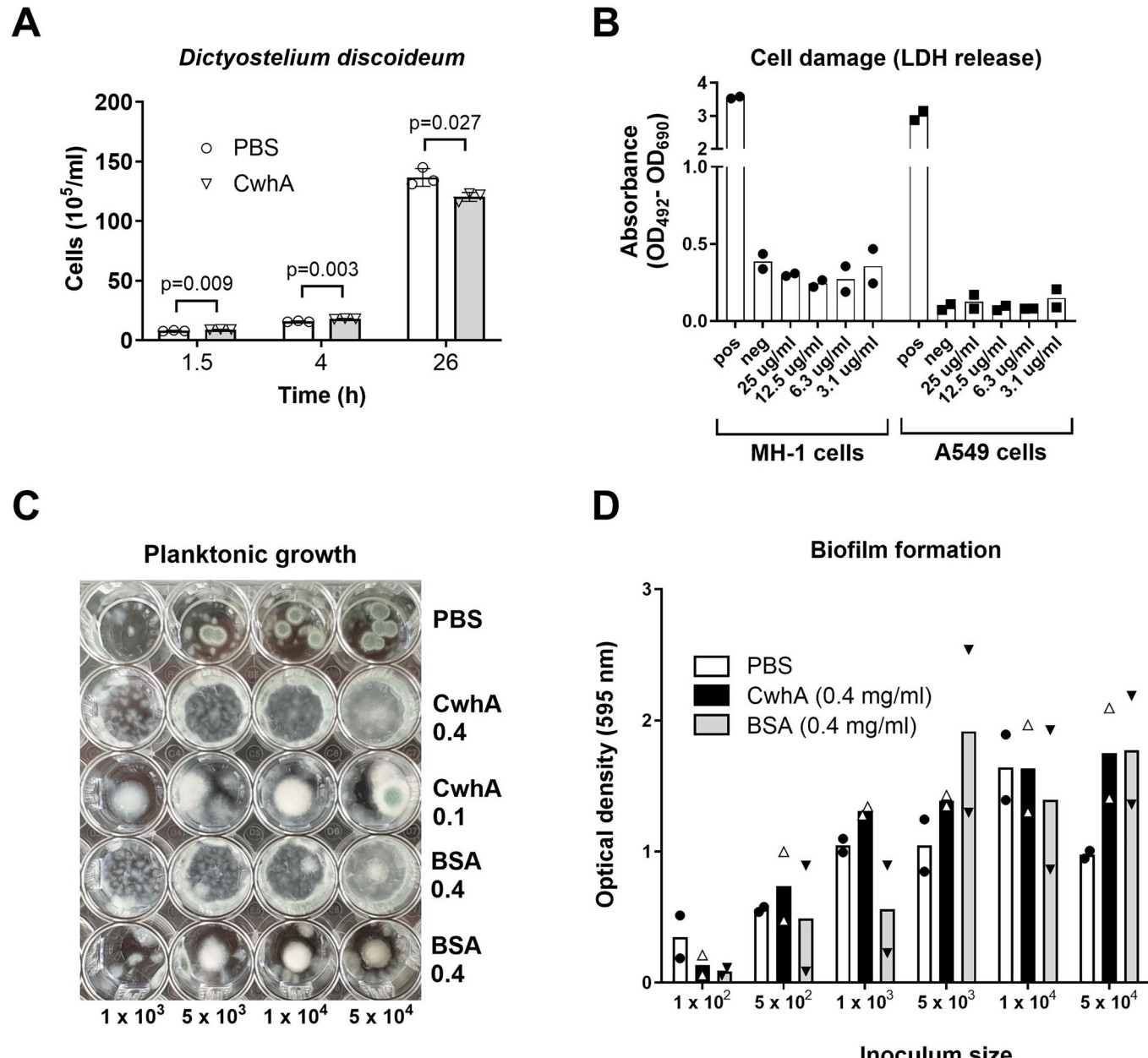

**Figure 6. Effect of CwhA on amoeba, host cells, and fungi.**

(A) Cell numbers of *Dictyostelium discoideum* in HL5 media following different incubation times with 130 μg/ml CwhA or PBS; $n = 3$ biological replicates. Data were analysed by two-way ANOVA (no significant impact of treatment) and per time point by two-tailed unpaired *t*-test (*p* values indicated in the graph). (B) Cell damage determined by lactate dehydrogenase (LDH) release of murine alveolar macrophages (MH-S) and human pulmonary epithelial cells (A549) after treatment with CwhA (3.1–25 μg/ml); $n = 2$ biological replicates. (C, D) Effect of CwhA on *A. fumigatus* CEA10 during planktonic growth (C) or (D) on biofilm formation. *A. fumigatus* spores ($1 \times 10^2$/ml to $5 \times 10^4$/ml) were inoculated in AMM supplemented with PBS, CwhA or Bovine serum albumine (BSA) at 0.4 and 0.1 mg/ml and incubated at 37 °C for 48 h either shaking (300 rpm) (C) or non-shaking (D) for adherent growth. (C) Single experiment. (D) $n = 2$ biological replicates. Data information: Data in (A) were presented as a scatter plot with mean (bar) ± SD (error bars). Data in (B, D) are shown as data points with a mean (bar). Data in (A) were analysed by two-way ANOVA (no significant impact of treatment) and per time point by two-tailed unpaired *t*-test (*p* values indicated in the graph). Source data are available online for this figure.

mutanolysin and CwhA. The cleavage products were separated by UHPLC and analysed by MS. When the digestion was performed with mutanolysin alone, both strains showed the expected muropeptides (Fig. 7A-I,B-I) (Atrih et al, 1999; Do et al, 2020). For *S. aureus*, these were mostly disaccharides (DS) with a stem peptide of five amino acids (pentapeptide), often harboring a

pentaglycine bridge. Several stem peptides can be cross-linked indirectly *via* these pentaglycine bridges, resulting in multimeric muropeptides (Figs. 7A-I and EV4). When *S. aureus* PG was treated with CwhA alone, the enzyme hydrolyzed a bond within the stem peptide, thereby segregating most of the peptide part of PG from the glycan strands (Fig. 7A-II). The resulting masses matched

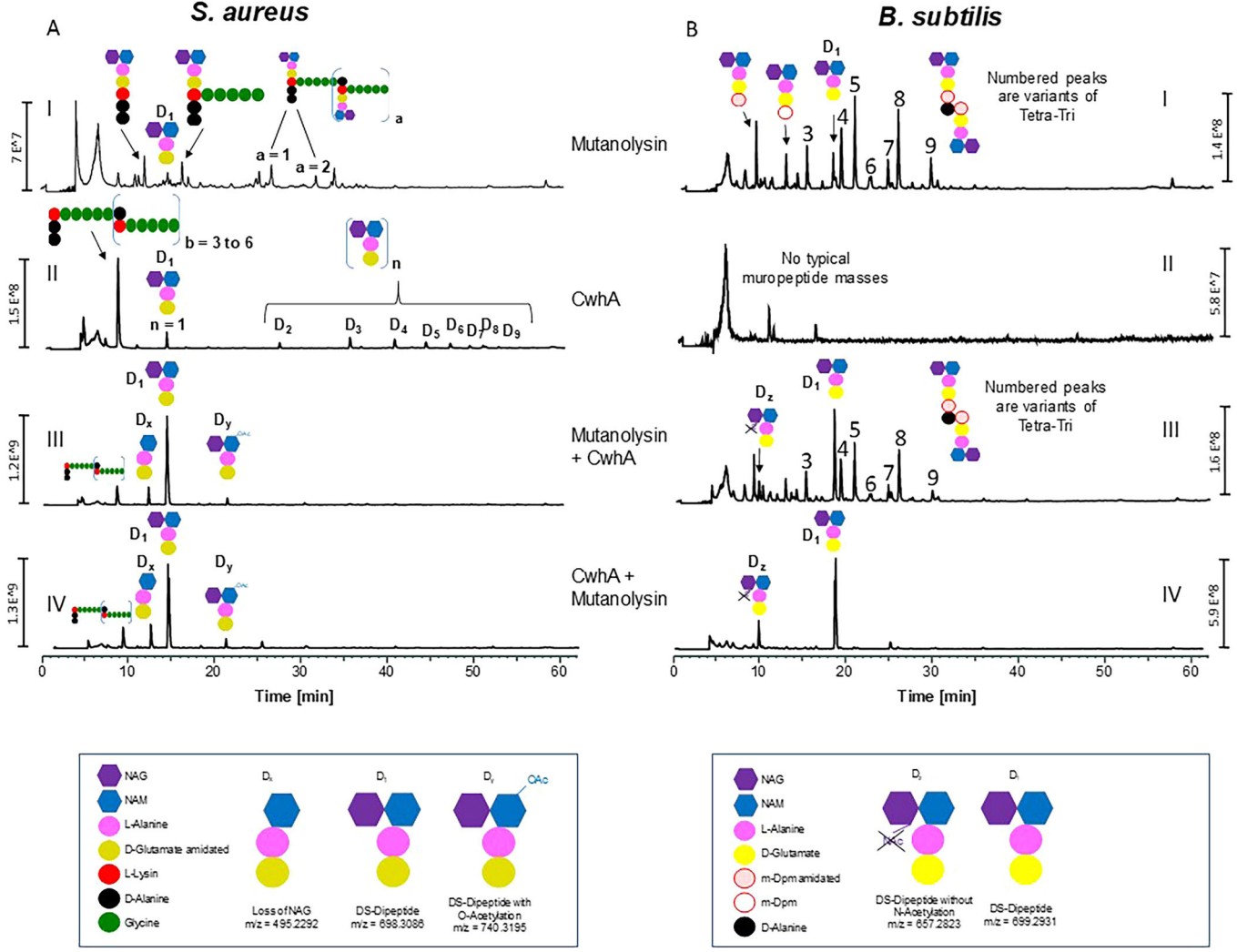

**Figure 7.** Muropeptide profile of *S. aureus* SA113 and *B. subtilis* (Ehrenberg 1835) obtained by UPLC/MS after treatment with mutanolysin or/and CwhA.

Masses of peaks are shown in Appendix Table S3. Numbered peaks are variants of Tetra-Tri. (**A**) Analysis of S. aureus PG cleavage. (**B**) Analysis of B. subtilis PG cleavage.

with peptide residues of four to seven subunits cross-linked *via* the pentaglycine bridge. The *m/z*-values observed in the other peaks belonged to glycan strands of one to nine DS-units that still contained one dipeptide (L-alanine - amidated D-glutamate) per *N*-acetylmuramic acid (NAM) residue. Therefore, the cleavage site of CwhA must be located between the second and the third amino acid of the stem peptide (in *S. aureus* this is amidated D-glutamate and L-lysine, respectively). When digested with both enzymes, mutanolysin and CwhA, the entire macromolecule was degraded into DS-dipeptides, the smallest PG cleavage product (Fig. 7A-III,A-IV). Three different variants of DS-dipeptides were observed: (i) unmodified DS-dipeptide is the most abundant form, (ii) DS-dipeptide without *N*-acetylglucosamine (NAG), and (iii) DS-dipeptide with an additional *O*-acetylation known to occur at position six of NAM [39,40] (Fig. 7A-III, a–c). The unmodified DS-dipeptide was also found in PG treated with mutanolysin alone (Fig. 7A-I), but its amount increased by a factor of 165 when CwhA was added (Fig. 7A-III,A-IV).

In contrast to *S. aureus*, *B. subtilis* contains mDpm at position three of the stem peptide. Within mature PG, the stem peptides are shortened to three or four amino acids by native enzymes, resulting in the formation of tetra- and tripeptides. Most muropeptides released by mutanolysin were variants of tetra-tri-dimers with various chemical modifications, such as amidation of mDAP and *N*-deacetylation of NAM (Fig. 7B-I) (Atrih et al, 1999). Treatment of *B. subtilis* PG with CwhA alone did not release any detectable muropeptides (Fig. 7B-II). However, when CwhA was added to PG that had been pre-treated with mutanolysin, the amount of DS-dipeptide increased while, concomitantly, the other muropeptides decreased (Fig. 7, compare panel B-III with panel B-I). When PG was initially treated with CwhA and subsequently with mutanolysin (resulting *de facto* in a prolonged CwhA treatment), even *B. subtilis* PG was completely digested into DS-dipeptides, as shown in Fig. 7, panel B-IV. Therefore, CwhA also accepts amidated mDpm-containing PG as substrate, not only the one with L-lysine. Unlike for *S. aureus*, the released peptide part was not detectable for *B.*

*subtilis*. For the most part, in *B. subtilis*, PG only two adjacent stem peptides are cross-linked without an interpeptide bridge, in contrast to three to seven stem peptides in *S. aureus*. This results in molecules of a very small size that do not facilitate retention on the column used.

Taken together, our data indicate that the enzyme CwhA is a fungal endopeptidase hydrolyzing the peptide bond between amino acids two and three of the PG stem peptides. It thereby seems to prefer L-lysine over mDpm on position three. However, with prolonged treatment, even PG containing amidated mDpm was degraded into DS-dipeptides, potentially explaining the lytic band detected in the zymogram gels (Fig. 2B). Further cleavage experiments demonstrated that CwhA in combination with mutanolysin efficiently degraded PG of two other Gram-positive bacterial strains containing L-lysine instead of mDpm, giving rise to a major peak for the DS-dipeptide (Appendix Fig. S1). One of the tested bacteria was an MRSA strain with the same PG structure as *S. aureus* Newman, the other one was *S. pneumoniae* (Klein 1884) that contains alanyl-serine cross-bridges or is directly cross-linked (Garcia-Bustos et al, 1987; Schleifer and Kandler, 1972). *B. subtilis* was retested and showed a similar muropeptide profile as PG treated with mutanolysin alone, and therefore, no activity of CwhA could be detected, most likely because the incubation time was shorter than when mutanolysin was added after treatment with CwhA. These data suggest that CwhA is a broad-spectrum enzyme that efficiently degrades the PG of various Gram-positive bacteria containing L-lysine at position 3 of the stem peptide, leading to high accumulation of the smallest PG cleavage product, DS-dipeptide. Thus, variations in the stem peptides explain the observed differences in CwhA efficacy against different bacterial species.

### CwhA-mediated PG cleavage induces cytokine responses

While CwhA itself did not damage host cells (Fig. 6B) and deletion of *cwhA* did not affect fungal survival if challenged with human macrophages (Fig. EV5), we hypothesized that CwhA activity might indirectly impact immune cells by processing of bacterial cell walls, leading to release of ligands for pattern recognition receptors. Although PG was described as a TLR2 ligand in early studies, subsequent investigations questioned this, but provided evidence that PG cleavage products stimulate NOD1 and NOD2 (Travassos et al, 2004) and were reviewed in (Wolf and Underhill, 2018).

We therefore measured IL-1β and TNF-α in supernatants of human PBMCs after stimulation with intact or digested PG. We confirmed cytokine induction by a combination of muramyl dipeptides (MDP) and heat-killed conidia, and furthermore observed a significant increase of both cytokines after stimulation with CwhA-degraded, but not intact, PG (Fig. 8A,B). Thus, in vivo bacterial cleavage by CwhA might affect immune responses and IPA severity.

## Discussion

As an environmental, saprophytic fungus, *A. fumigatus* inhabits ecological niches with high microbial diversity and cell density, leading to a strong interspecies communication. It is thus not surprising that, in addition to metabolic versatility, antimicrobial

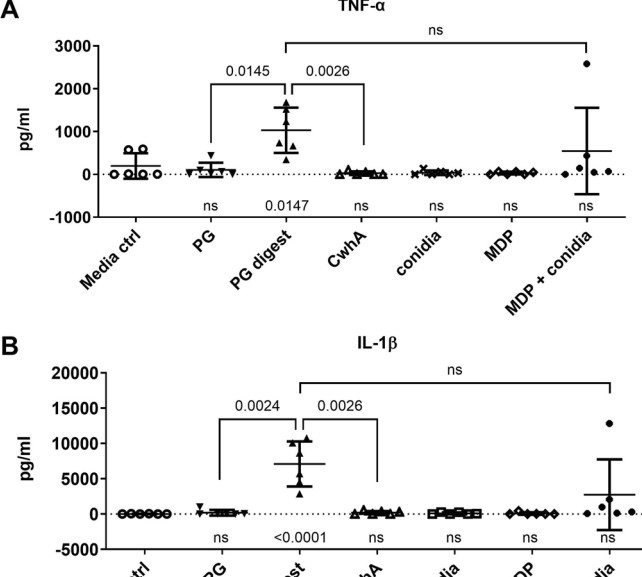

**Figure 8. PG cleaved by CwhA leads to cytokine induction of human PBMCs.**

(A, B) *Aspergillus*-MDP- induced TNF-α (A) and IL-1β (B) in culture supernatants of PMBCs ($5 \times 10^5$) of healthy volunteers ($n = 6$ biological replicates). Media ctrl: Medium (RPMI 1640+ with 10% human serum) only; PG: *S. aureus* PG only; PG digest: *S. aureus* PG + 20 μg CwhA; CwhA: 20 μg CwhA; conidia: $1 \times 10^7$/ml heat-killed conidia of *A. fumigatus* CEA10; MDP: 10 μg/ml MDP; MDP + conidia: 10 μg/ml MDP and $1 \times 10^7$/ml heat-killed conidia of *A. fumigatus* CEA10. Data information: Data in (A, B) were presented as a scatter dot plot with mean ± SD. The data were analysed by (i) Ordinary one-way ANOVA and Dunnett's multiple comparisons test to compare all groups with the control; results (*p* value or ns not significant) are indicated in the graph below the data points. (ii) Pairwise comparison of selected groups was performed using a two-tailed unpaired *t*-test as indicated in the graph. Source data are available online for this figure.

mechanisms evolved in fungi to promote survival in highly competitive environments (Macheleidt et al, 2016; Scharf et al, 2016). For example, filamentous fungi produce an enormous structural diversity of secondary metabolites with a broad spectrum of activities and functions, e.g., for communication with other microorganisms or combating bacterial competitors (Macheleidt et al, 2016).

Here, we found a special protein that has similar functions. It is the previously uncharacterized fungal protein B0YAY0, a secreted endopeptidase that specifically cleaves distinct bonds in the stem peptide of bacterial PG. Due to its activity, and similarity of the functional domains with bacterial cell wall hydrolases, we renamed the protein cell wall hydrolase A (CwhA). Bacterial cell wall hydrolases are essential for cell division, cell wall recycling, and differentiation. Therefore, bacteria produce a wide range of these enzymes that differ in PG-binding and cleavage sites and their substrate specificity (Do et al, 2020). Some murein hydrolases, so-called bacteriolysins, are also used by bacteria to eliminate other bacterial strains that share the same ecological niche and compete for nutrients and other limiting factors (Wittekind and Schuch, 2016). A well-characterized example is lysostaphin, which is

produced by *Staphylococcus simulans* biovar *staphylolyticus*. This metalloenzyme cleaves pentaglycine cross-bridges in PG of other *Staphylococcus* strains to eliminate competitors (Bastos et al, 2010). Cell wall hydrolases are also produced by animals, including humans: Lysozyme, which we used in our study as a positive control, is highly abundant in various body fluids and involved in host defense against bacteria by cleavage of peptidoglycan and subsequent bacterial lysis (Ragland and Criss, 2017). The conservation of functional lysozymes throughout the animal kingdom underscores their value in combating bacteria (Callewaert and Michiels, 2010); it is thus not surprising that fungi like *A. fumigatus* produce proteins with similar functions to defend the habitat against bacteria.

CwhA is not produced under standard laboratory culture conditions, which explains why its function was not discovered earlier. The transcription of *cwhA* was, however, upregulated during coculture with some bacteria. Physical contact with bacteria also induces production of secondary metabolites with antibacterial potential (Fischer et al, 2018; Netzker et al, 2015; Schroeckh et al, 2009). This suggests a general fungal strategy to produce antimicrobial metabolites and proteins only in the presence of other microorganisms when these compounds are beneficial. Of note, although increased *cwhA* transcription during coculture with *S. aureus* was observed, the level of induction was only moderate. Likely, the transcriptional upregulation observed does not lead to sufficient production of secreted CwhA to kill a substantial number of bacteria in the assays employed here, thereby explaining why only the overexpression strain but not *A. fumigatus* WT showed substantial bacterial killing in the co-cultivation experiments.

This is in contrast to the local induction of the secreted protein in vivo during IPA in mice. In *Aspergillus fumigatus* Af293 *cwhA* (Afu800360) is located in a "supercluster" (Afu8g00100–Afu8g00720) on chromosome 8. This supercluster region includes gene clusters responsible for the biosynthesis of fumitremorgin, fumagillin and pseurotin A (Wiemann et al, 2013). The regulation of genes in this chromosomal region is rather complex and includes a number of different transcription factors (Dhingra et al, 2013; Wiemann et al, 2013; Yu et al, 2018). For CwhA, such a regulator has not been identified yet that explains its apparently higher expression in vivo (mouse lungs) than in vitro in mono culture of *A. fumigatus*. Although we did not observe an upregulation of *cwhA* under hypoxia or in contact with intact lung epithelial cells, *cwhA* gene expression was increased at a late time point during infection of A549 cells. This coincided with microscopically observable host cell damage, the release of intracellular LDH, and supernatants of cells infected with the fungus also led to upregulation of *cwhA*. This might indicate danger-associated molecular patterns (DAMPs) as one signal that is involved in *cwhA* regulation. The failure of lysates generated from intact cells to induce expression does not directly contradict this hypothesis, but points towards inducible DAMPs (Yatim et al, 2017) as possible signals. This hypothesis, however, remains to be further tested, and it is also possible that not a single factor but a combination of stimuli drives expression in vivo.

Furthermore, we also identified two other uncharacterized fungal proteins, B0Y269 and B0Y9E0, which were highly abundant in BAL samples of mice with invasive aspergillosis, and shared common motifs with CwhA (Machata et al, 2020). Like CwhA, these proteins appear not to be produced in vitro, but the gene encoding B0Y269 was highly upregulated in vivo (McDonagh et al,

2008). These common features with CwhA are a hint that the proteins might have similar functional roles and could work in concerted action with CwhA, possibly showing different substrate specificities, with a preference towards other bacteria. Further work needs to be done to investigate these proteins in more detail and to elucidate their functions for fungal physiology.

Since we found no indication that CwhA is cytotoxic or that it is important for the interaction with macrophages, it seems unlikely that the protein directly affects virulence. CwhA might, however, influence interactions between *A. fumigatus* and bacteria during polymicrobial pulmonary colonization or infection. In contrast to other body niches, such as the gut and upper respiratory tract, that are characterized by development of a relatively stable microbiota resilient to colonization by external microbes, the lung microbiome is highly dynamic. Microbes reaching the lower airways commonly originate from the environment, oral cavity or upper respiratory tract, and are usually cleared rapidly by innate and adaptive immunity (Natalini et al, 2023). This results in low microbial abundance and immunological priming that affects later responses to respiratory pathogens (Wu et al, 2021). While it seems questionable that the low-density, variable lung microbiota mediates colonization resistance to the extent demonstrated for the gut microbiota, it can affect colonization by other microbes through host immunity and inter-microbial interactions. Interactions between members of the microbiota shape the microbial community in the upper respiratory tract (Man et al, 2017), and are likely also relevant in pathophysiological settings associated with higher microbial abundance in the lung, for example, cystic fibrosis (O'Brien and Fothergill, 2017). However, little is known about microbial interactions in the lung, especially related to fungi. The role of allergic bronchopulmonary aspergillosis in CF patients has been firmly established (Carsin et al, 2017), but the effects of fungal colonization without infection on CF remain unclear. By reducing *S. aureus* colonization, *A. fumigatus* might promote the growth of Gram-negative bacteria such as *P. aeruginosa* that are not impacted by CwhA, and thereby contribute to disease development. Conversely, bacteria could impact fungal colonization, and inter-kingdom interactions not only affect colonization levels but also expression of virulence factors (Rozaliyani et al, 2023).

Another effect that accompanies CwhA-mediated bacteriolysis is the accumulation of increased levels of MDP due to the cleavage of bacterial peptidoglycan. The increased cytokine levels upon treatment with PG pre-digested beforehand with CwhA confirmed our hypothesis that the release of CwhA during infection could affect the immune stimulation in the host. This finding is particularly intriguing from the immunological point of view, as Gresnigt et al showed that NOD2 stimulation by a combination of *A. fumigatus* conidia and MDP suppresses fungal killing in human monocyte-derived macrophages (MDMs) or monocytes and leads to increased cytokine production (Gresnigt et al, 2018). Furthermore, NOD2 receptor deficiency renders mice more resistant to IPA, and risk patients with NOD2 polymorphisms are less predisposed to IP (Gresnigt et al, 2018). Bacterial Muramyl-dipeptide (MDP) is a known NOD2 agonist and can synergistically increase *Aspergillus*-induced cytokine levels (Li et al, 2012; Zhang et al, 2008). However, it still remains unclear how NOD2 is activated by *A. fumigatus*. Our data suggests that CwhA-mediated cleavage of PG into muropeptides that serve as NOD2 ligands could contribute to the effects of IPA in an in vivo setting where Gram-

positive bacteria are present. The production and activity of CwhA by *A. fumigatus* in the lung of a host who harbors microbiota and is constantly exposed to external bacteria could explain the MDP-mediated stimulation and thus the relevance and importance of the NOD2 receptor for *Aspergillus* infection. However, if CwhA-generated PG cleavage products are indeed sensed by NOD2, it remains to be investigated, and further studies are necessary to gain a better insight into the validity and mechanism of the CwhA contribution to host immune responses and fungal survival in the host.

In summary, CwhA is the first secreted antibacterial protein identified in *A. fumigatus*, complementing other fungal antibacterial factors such as secondary metabolites and defensins. As an endopeptidase, it cleaves PG of Gram-positive bacteria, leading to bacterial lysis and the release of PG cleavage products that stimulate cytokine production of human immune cells. Its expression in the lungs of mice infected with *A. fumigatus* can be partially explained by the presence of bacteria, but is likely triggered by complex stimuli. Together with other fungal factors that influence bacterial growth, it might impact the microbiome in both environmental and clinical niches inhabited by *A. fumigatus*.

# Methods

### Reagents and tools table

| Reagent/resource | Reference or source | Identifier or catalog number |
|---|---|---|
| *Aspergillus* minimal medium (AMM) | (Brakhage and Van den Brulle, 1995) | N/A |
| Recombinant CwhA protein | this study | N/A |
| **Experimental models** | | |
| MRSA strains | see Appendix Table S4 | clinical isolates |
| *A. fumigatus* CEA10 | (Monod et al, 1993) | clinical isolate |
| *A. fumigatus* CEA17 Δ*akuB*^KU80 | (da Silva Ferreira et al, 2006) | N/A |
| CEA17 Δ*akuB*^KU80Δ*cwhA* | This study | N/A |
| CEA17 Δ*akuB*^KU80::P*gpdA::cwhA* | This study | N/A |
| *A. fumigatus* strain C3 | (Brock et al, 2008) | N/A |
| *Staphylococcus aureus* | Jena Microbial Resource Collection (JRMC) | Newman |
| *Staphylococcus aureus* | JRMC | SA113 |
| *Staphylococcus aureus* | (Balwit et al, 1994; Kahl et al, 2000) | 6850/pALC1743 |
| *Bacillus subtilis* | JRMC | DSM3256 |
| *Streptococcus pneumoniae* | JRMC | 6303 |
| *S. epidermidis* | JRMC | IMET10845 |
| *Klebsiella pneumoniae* | JRMC | 700603 |
| *Pseudomonas aeruginosa* | JRMC | PAO1 |

| Reagent/resource | Reference or source | Identifier or catalog number |
|---|---|---|
| *Enterococcus faecalis* | This study | Clinical blood isolate BK014967, kindly provided by Steffen Höring, University Hospital Jena |
| *E. coli* DH5α | Invitrogen (Thermo Fisher) | EC0112 |
| *E.coli* KRX | Promega | L3002 |
| A549 cells, human | ATTC | CCL-185 |
| MH-S cells, murine | ATTC | CRL-2019 |
| *D. discoideum* AX2 | (Bloomfield et al, 2008) | Kindly provided by Thomas Winckler, Friedrich Schiller University Jena |
| Adherent human monocyte-derived macrophages | (Sprenger et al, 2020) | N/A |
| PBMCs | this study | N/A |
| **Recombinant DNA** | | |
| pSK275 | (Szewczyk et al, 2006) | N/A |
| pSK379 | (Szewczyk and Krappmann, 2010) | N/A |
| pRSETb | Addgene | pRSET T7 expression vector |
| **Antibodies** | | |
| CwhA^rec mouse monoclonal antibody 2E6D | Custom-made for this study by LeukoCom GmbH, Essen, Germany | N/A |
| Anti-Mouse IgG (H + L), HRP Conjugate | Promega | W4021 |
| **Oligonucleotides and other sequence-based reagents** | | |
| PCR primers | This study | see Appendix Table S5 |
| **Chemicals, enzymes and other reagents** | | |
| Pierce™ ECL western blotting substrate | Thermo Scientific | 32106 |
| Immobilon-P Membran, PVDF | Merck Millipore | IPVH00010 |
| Nystatin | Merck | 475914 |
| Pyrithiamine hydrobromide | Sigma-Aldrich | P0256-1MG |
| Phusion High Fidelity DNA Polymerase | NEB | M0530S |
| XhoI | NEB | R0146S |
| SacI | NEB | R3156S |
| PmeI | NEB | R0560S |
| Rhamnose | Sigma-Aldrich | R3875 |
| DNase | Roche | 04716728001 |
| RNase | Roche | 11119915001 |
| Trypsin | Sigma-Aldrich | T4799 |
| Lysozyme | Sigma-Aldrich | L6876 |
| Bovine serum albumin | Serva | 111930.01 |

| Reagent/resource | Reference or source | Identifier or catalog number |
| --- | --- | --- |
| Sodium dihydrogen phosphate | Sigma-Aldrich | 5438400100 |
| Mutanolysin | Sigma-Aldrich | M9901 |
| PBS | Gibco | 11503387 |
| Fetal Calf serum | Bio&SELL | FBS.SAM.0500 |
| RL buffer | Roboklon | E0310-01 |
| HL5 media, including glucose | Formedium | HLG0101 |
| DMEM | Gibco | 11966025 |
| RPMI 1640 | Gibco | 11530586 |
| Penicillin/streptomycin solution | Gibco | 11548876 |
| Superscript III Reverse Transcriptase | Invitrogen | 18080093 |
| LC-MS grade water | VWR | MERC1.15333. |
| LC-MS grade methanol | VWR | 92498.320 |
| Lymphocyte separation medium density 1.077 g/ml | Capricorn Scientific | LSM-A |
| Muramyl- dipeptide | Invivogen | Ilrl-mdp |
| **Software** | | |
| GraphPad Prism | https://www.graphpad.com/ | GraphPad version 9.12 |
| Image J | https://imagej.nih.gov/ij/index.html | N/A |
| SCiLS Lab software | Bruker Daltonics | Version 2020a Premium 3D |
| **Other** | | |
| Conductive ITO slides | Bruker Daltonics | 8259387 |
| SunCollect | SunChrom GmbH | N/A |
| SunDigest | SunChrom GmbH | N/A |
| UltrafleXtreme mass spectrometer | Bruker Daltonics | N/A |
| Pannoramic DESK | Sysmex | N/A |
| CASY Cell Counter | Roche Innovatis | N/A |
| CASY Cell Counter | Omni Life Science | N/A |
| Miracloth | Merck Millipore | 475855 |
| Tecan Infinite M200 Pro | Tecan | N/A |
| French press Gaulin | APV | N/A |
| Emulsiflex C5 | ATA Scientific instruments | N/A |
| IMAC XK25/20 chromatography column | Pharmacia | 28-9889-48 |
| Ni-Sepharose FF | Cytiva | 17-5318-02 |
| Resource S chromatography column | Cytiva | 17-1180-01 |
| IMAC XK16/20 CoNTA column | Cytiva | 28-9889-37 |

| Reagent/resource | Reference or source | Identifier or catalog number |
| --- | --- | --- |
| ÄKTA Pure 25 | GE Health Life Sciences | N/A |
| Amicon centrifugal units 10 kDa pore size | Millipore | UFC9010 Millipore |
| Pierce™ Bradford Protein Assay Kit | Thermo Scientific™ | 23200 |
| Heating block | ThermoStat plus, Eppendorf | N/A |
| Water bath sonicator | Sonorex, Bandelin | N/A |
| Thermo QExactive Plus Orbitrap | Thermo Fisher | N/A |
| Precellysis24 homogenizer | Bertin Technologies | N/A |
| Automated live cell imaging microscope | Zeiss | Celldiscoverer7 |
| Hypoxystation | Don Whitley Scientific | H35 |
| Ham's F-12K (Kaighn's) Medium | Gibco | 11580556 |
| ACQUITY UHPLC column | Waters | 186002350 |
| UHPLC Agilent 1290 | Agilent | N/A |
| Microplate reader | ClarioSTAR plus | N/A |
| Glass beads 0.5 mm | Biospec. Products | 11079105 |
| GeneMatrix Universal RNA Purification Kit | Roboklon | N/A |
| Nanodrop ND-1000 | Thermo Fisher | N/A |
| RNA 600 Nano Kit | Agilent Technologies | 5067-1511-7643684 |
| Bioanalyzer 2100 | Agilent Technologies | N/A |
| 2xGo Taq qPCR Mastermix | Promega | A6001 |
| S-Monovette® EDTA K3E, 9 ml | Sarstedt | 02.1066.001 |
| ELISA kit for IL-1β | Invitrogen | 88-7261-86 |
| ELISA kit for TNF-α | Invitrogen | 88-7346-22 |
| 96-well Microplates, PS, F-Bottom | Greiner | 655906 |
| Falcon 96-well flat-bottom plates | Corning | 351172 |
| 96-well flat-bottom plates | TPP | 92696 |
| 24-well flat-bottom plates | TPP | 92424 |
| Six-well plates | TPP | 92406 |
| Microscope | Zeiss | SteREO Discovery V8 |

## Lung biopsy and MALDI-IMS

Murine lungs were obtained from immunocompromised animals infected intranasally with *A. fumigatus* as part of a previous study (Machata et al, 2020). Lungs were stored in 4% formalin (Histofix), samples were embedded in paraffin, cut into 5 μm sections, and transferred onto conductive ITO slides (Bruker Daltonics). About

1 μl of 200 ng/ml recombinant CwhA was spotted next to the tissue sections, and served as a control for the tryptic digestion. The sample preparation was performed as described previously (Hoffmann et al, 2019). Shortly, the sections underwent deparaffinization, rehydration, pH-conditioning, and antigen retrieval. For trypsin deposition, the SunCollect (SunChrom GmbH) was used. The tryptic digestion was performed using the digestion chamber SunDigest (Sunchrom GmbH) in "smart mode", using the following protocol: number of steps: 2; step 1: 900 s, 0% fan speed, base temperature: 50 °C, cover temperature: 45 °C; step 2: 6300 s, 2% fan speed, base temperature: 50 °C, cover temperature: 45 °C. The matrix used for all sections consisted of 30 mg/ml 2,5-dihydroxybenzoic acid (DHB) in 50% acetonitrile and 0.2% trifluoroacetic acid. Matrix application was performed with the SunCollect with the following parameters: five layers; flow rate for layer 1: 10 μl/min; flow rate for layers 2 to 5: 35 μl/min; speed x: low 3, speed y: medium 1, vial distance: 2 mm, Z position: 29 mm. MALDI-TOF – IMS data acquisition was performed using the UltrafleXtreme mass spectrometer (Bruker Daltonics). The measurements were carried out in positive ion reflective mode, with 50 μm pixel size, medium laser beam size, 200 shots per position, in a mass range of $m/z$ 500 to 4000. The Bruker peptide calibration standard was used for external mass spectrometer calibration. After MALDI-IMS measurements, the sections were washed twice with 80% ethanol and PAS (periodic acid-Schiff) -stained for histopathological annotation. The slides were scanned using the Pannoramic DESK (Sysmex). For data and imaging analysis, the SCiLS Lab software (Bruker Daltonics) was used. All data were TIC-normalized. For in silico digestion, the ProteinProspector MS-Digest tool was used (http://prospector.ucsf.edu).

## Microorganisms, media, and cultivation

*A. fumigatus* wild-type strains CEA10 (FGSC A1163; wildtype used for most experiments unless otherwise stated) and its derivatives CEA17 $\Delta akuB^{KU80}$ (da Silva Ferreira et al, 2006), CEA17 $\Delta akuB^{KU80}\Delta cwhA$ (this study), and CEA17 $\Delta akuB^{KU80}$::P$gpdA$::$cwhA$ (this study) were used for in vitro experiments. The bioluminescent *A. fumigatus* strain C3 (based on CGS144.89, the wild-type origin of CEA10) was kindly provided by Matthias Brock (University of Nottingham, UK) and used for murine infection experiments in a previous study (Brock et al, 2008), which provided the lung tissue used here. Fungal spores were grown at 37 °C on *Aspergillus* minimal medium (AMM) agar for three days (Brakhage and Van den Brulle, 1995). When required, the medium was supplemented with 1 μg/ml pyrithiamine hydrobromide (Sigma-Aldrich). Conidia were harvested from agar plates with water and counted using the CASY Cell Counter (Roche Innovatis). For liquid cultures, *A. fumigatus* conidia were introduced into the liquid medium specified below at a final concentration of $10^6$ conidia/ml and cultured at 37 °C on a rotary shaker set to 200 rpm for the specified durations.

Fungal mycelium was harvested with Miracloth (Merck Millipore) from liquid cultures of *A. fumigatus* strains grown overnight in AMM medium at 37 °C, after washing with water or with sterile PBS when required. *Staphylococcus aureus* (Newman and SA113), *Bacillus subtilis* (DSM3256), *Streptococcus pneumoniae* (ATCC 6303), *S. epidermidis* (IMET10845), *Klebsiella pneumoniae* (ATCC 700603), *Pseudomonas aeruginosa* (PAO1), and *Enterococcus faecalis* (clinical blood isolate BK014967, kindly provided by

Steffen Höring, University Hospital Jena) wild-type strains were used for peptidoglycan isolation, lysis assays and/or co-cultivation experiments, culture details are provided below. Additional *S. aureus* strains (multiresistant, MRSA) used to test the effect of CwhA are listed in the Appendix Table S4. The GFP-expressing *Staphylococcus aureus* strain (6850/pALC1743) (Balwit et al, 1994; Kahl et al, 2000) was used to determine the lytic activity of recombinant CwhA. Bacterial cultures were grown in LB medium or on LB agar plates. In survival assays after co-cultivation with *A. fumigatus*, 0.1 mg/ml nystatin (Merck) was added to LB agar plates to prevent fungal growth.

## Genetic manipulations and purification of recombinant protein

A list of oligonucleotides used in this study is provided in the Appendix Table S5. The generation of PCR fragments for all genetic manipulations was carried out using the Phusion High Fidelity DNA Polymerase (NEB). *A. fumigatus* deletion and overexpression strains were generated using CEA17 $\Delta akuB^{KU80}$ as the parental strain. The AFUB_086210 (*cwhA*) deletion (resulting in strain CEA17 $\Delta akuB^{KU80}\Delta cwhA$) was performed by replacing the gene with a pyrithiamine resistance cassette generated with ptrA_for_II and ptrA_rev_II primers from plasmid pSK275 (Szewczyk et al, 2006) using 5′ and 3′ flanking regions of AFUB_086210, generated with primers Del_p60A_F1 and Del_p60A_R2, and Del_p60A_F3 and Del_p60A_R4, respectively. The deletion was achieved by homologous recombination following transformation of protoplasts as previously described (Weidner et al, 1998). Correct deletion was confirmed by PCR and by Southern blot: The gene-specific DNA probe was generated by PCR using the primers Del_p60A_F1 and Del_p60A_R4. Genomic DNA isolated from the wildtype and the mutant strains, was digested with the restriction enzymes XhoI and SacI (NEB). The over-expressing strain CEA17 $\Delta akuB^{KU80}$::P$gpdA$::$cwhA$ was constructed using the vector pSK379 (Szewczyk and Krappmann, 2010) (kindly provided by Sven Krappmann, University of Erlangen) containing a pyrithiamine cassette and the P$gpdA$ promoter. The *cwhA* gene was amplified with primers Expr_p60A_for and Expr_p60A_rev, using genomic DNA derived from *A. fumigatus* CEA10 as template, and integrated into the plasmid linearized with PmeI (NEB) restriction enzyme. The CEA17 $\Delta akuB^{KU80}$::P$gpdA$::$cwhA$ was obtained by homologous recombination following the transformation of protoplasts as previously (Weidner et al, 1998). For the generation of the recombinant CwhA protein, the AFUB_086210 gene was amplified from genomic DNA of *A. fumigatus* CEA10 using o_p60AFor and o_p60ARev primers. o_p60AFor binds after the sequence encoding the signal peptide, and therefore after the intron in gene AFUB_086210. The amplified DNA was cloned into the plasmid pRSETb (Addgene) at its multiple cloning site C-terminal of a His-tag sequence and transformed into *E. coli* DH5α (Invitrogen). Isolated plasmid from positive transformants was verified by sequencing and transformed into the *E. coli* KRX strain (Promega) for protein expression. Expression was carried out by fermentation in TY media at 25 °C with a Start-OD of 0.6 and induction by 0.1% rhamnose after 1 h, continued for 6 more hours. Cells were collected by centrifugation at 4 °C, 10,000 × *g* for 45 min and lysed by French press followed by cell disruption using the homogenizer Emulsiflex C5 (ATA Scientific instruments) after

resuspension of cell in Ni-NTA buffer (50 mM sodium phosphate, by another 300 mM sodium chloride, 10 mM imidazole). Protein purification was performed using immobilized metal chromatography (IMAC XK25/20 chromatography column loaded with 25 ml Ni-Sepharose FF, Cytiva), followed by Resource S ion exchange chromatography (Cytiva) and another immobilized metal chromatography (IMAC XK16/20 chromatography column loaded with 10 ml Ni- Sepharose FF, stripped and reloaded with cobalt ions) on the ÄKTA Pure 25 (GE Health Life Sciences) purification system. The protein was concentrated using Amicon centrifugal units with 10 kDa pore size (Millipore), the final protein concentration was evaluated by Bradford assay and protein purity was assessed by SDS-PAGE. Protein with >90% purity was kept at 4 °C for short- term storage or supplemented with 10% glycerol at −20 °C for long-term storage.

## Preparation of *Aspergillus* proteins and Western blot

Fungal protein extracts were performed as previously described (Vödisch et al, 2011) with modifications. Harvested mycelium from AMM liquid cultures was washed with water, and the mycelium was dried using paper towels and immediately frozen in liquid nitrogen for further protein extraction. Frozen mycelium was ground in a chilled mortar in the presence of liquid nitrogen, and 500 mg of mycelia was transferred to a 2 ml tube for lysis with lysis buffer (50 mM Tris pH 7.0, 3% SDS (w/v), 0.05 mM EDTA). The tubes were incubated for 2 min at 42 °C and exposed to liquid nitrogen immediately after. This heat-shock procedure was repeated three times, samples were centrifuged for 5 min at 12,000 × $g$, 20 °C and supernatant was collected.

Isolation of secreted proteins was performed as previously described (Wartenberg et al, 2011). Culture supernatants were collected and centrifuged at 12,000 × $g$ for 30 min to remove cell debris or insoluble compounds. Proteins were precipitated by the addition of 10% (w/v) trichloroacetic acid to the culture supernatant and subsequent incubation for 12 h at 4 °C. The pellet was collected by centrifugation (45 min, 4000 × $g$ at 4 °C) and air-dried for 10 min at room temperature. The pellet was resuspended with a solubilization buffer (7 M urea, 20 mM Tris pH 7, 0.1% Triton (w/v)), and solubilized proteins were separated from non-soluble particles by collecting the supernatant after 10 min centrifugation (10,000 × $g$, 4 °C) and storage at −20 °C.

Samples containing 50 μg protein from cell extracts and supernatants were separated using 12.5% SDS-PAGE, transferred to PVDF membranes (Merck Millipore) and probed with anti-CwhA[rec] mouse monoclonal antibody 2E6D generated by an external service provider (LeukoCom GmbH, Essen, Germany). Detection was performed using horseradish peroxidase-coupled anti-mouse IgG antibody (Promega) and Pierce™ ECL western blotting substrate (Thermo Scientific) per the manufacturer's instructions.

## Peptidoglycan isolation for LC-MS analysis

Peptidoglycan was isolated from *Staphylococcus aureus*, MRSA 181, *Bacillus subtilis* and *Streptococcus pneumoniae* overnight cultures as previously described in detail (Kühner et al, 2014). Briefly, cells from 3 ml overnight cultures grown in LB medium were harvested by centrifugation at 4 °C and 10,000 × $g$ for 10 min. Cells were

resuspended in 1 ml 1 M sodium chloride and boiled at 100 °C for 20 min in a heating block (ThermoStat plus, Eppendorf). The suspension was centrifuged at 10,000 × $g$ for 5 min, cells were washed four times in ddH$_2$0 and the pellet was resuspended in 1 ml ddH$_2$0. The sample was transferred to a water bath sonicator (Sonorex, Bandelin) for 30 min, after which 500 μl of 0.1 M Tris/HCl, pH 6.8, containing 15 μg/ml DNase (Roche) and 60 μg/ml RNase (Roche) were added. The suspension was incubated for 60 min at 37 °C with agitation at 180 rpm, 500 μl of a 50 g/ml trypsin solution was added and the suspension was incubated for another 60 min under the same conditions. Enzymes were inactivated by boiling the suspension for 3 min at 100 °C, cells were spun down at 10,000 × $g$ for 5 min and washed once in 1 ml ddH$_2$0. The pellet was resuspended in 500 μl 1 M HCl and incubated for 4 h at 37 °C at 180 rpm. The suspension was centrifuged and washed with ddH$_2$0 until the pH increased to 5–6. For digestion, the isolated PG was centrifuged, resuspended in 12.5 mM sodium dihydrogen phosphate (pH 5.5) and the optical density at 600 nm (OD$_{600}$) was determined. Cell wall material was diluted to an OD$_{600}$ of 3.0 in the same buffer to a final volume of 200 μl, and 20 μl mutanolysin (5000 U/ml; Sigma-Aldrich) and/or 10 μl CwhA (4 mg/ml) or PBS were added. The digest was incubated at 37 °C, agitated at 150 rpm for 16 h, and the reaction was inactivated by boiling for 3 min at 100 °C. Samples were cooled on ice until the cleavage products were analysed by LC-MS.

## LC-MS analysis of cleavage products

Immediately prior to the analysis, muropeptides were reduced using sodium borohydride (Sigma-Aldrich), and pH was adjusted to 3 with formic acid. Subsequent UHPLC analysis was performed as previously described (Kühner et al., 2014), with instrument modifications. Muropeptide separation was performed with an ACQUITY UHPLC column (Waters) on an Agilent 1290 and a G42220 Binary Pump running in UHPLC mode with a maximum pressure limit of 1200 bar. Injection volume was 15 μl. The mobile phase consisted of LC-MS grade water (VWR) with 0.2% formic acid (Sigma-Aldrich; A) and LC-MS grade methanol (VWR) with 0.2% formic acid (B). The flow rate was 0.265 ml/min. For the initial 4 min, the mobile phase was held at 100% A and the flow was discarded, in order to eliminate salt contaminants. From 4.10 min, the gradient started with 3% B and continued to 30% B at 57 min. The column was washed for 5 min with 30% B, followed by a change to 100% A in 1 min, and an equilibration step for 6 min at 100% A. For MS analysis, a Thermo QExactive Plus Orbitrap equipped with a HESI ion source running in positive mode was used. For full scan, the resolution was 17.000 with an AGC target of 1e6 and max. IT of 150 ms.

## Zymogram gels

Peptidoglycan isolation from *S. aureus* and *B. subtilis* for zymogram gels was conducted as described before (Fukushima and Sekiguchi, 2016). Briefly, bacterial cultures were grown in LB medium overnight (37 °C, 180 rpm), cells were collected by centrifugation (6000 × $g$, 15 min), washed twice in PBS, and ruptured with glass beads using the Precellysis24 homogenizer (Bertin Technologies). Beads were removed by filtering, and the cell wall fraction was pelleted by centrifugation (20,000 × $g$, 30 min). The pellet was boiled in 8% SDS for 10 h, washed three times with

PBS, resuspended in PBS, and stored at 4 °C. Lytic activity of the CwhA protein was detected using the protocol of de Jonge et al (de Jonge et al, 1991). Briefly, proteins were resolved using 12.5% SDS polyacrylamide gels containing 10 µl/ml isolated cell wall suspension (*S. aureus* or *B. subtilis*). After gel electrophoresis, SDS was removed by washing the gels three times in 50 ml 25 mM Tris-HCl (pH 7) containing 1% of Triton X-100 within a period of 18 h at room temperature. Gels were stained in 20 ml of staining solution (1% methylene blue in 0.01% KOH) for 30 s and immediately destained in $H_2O$. The water in the washing bath was changed until a lytic band was clearly visible.

## Live cell imaging of GFP-expressing *S. aureus*

The GFP-expressing *S. aureus* strain (Balwit et al, 1994; Kahl et al, 2000) was harvested from LB agar plates, a homogenous suspension was prepared in 50 mM Tris buffer (pH 8) and diluted to an optical density of 1.0 at 600 nm. 100 µl of the suspension was transferred into single wells of a flat-bottom 96-well microplate (Greiner). After the addition of 5 µl PBS, BSA (bovine serum albumin, 2 mg/ml or 160 µg/ml), or CwhA (2 mg/ml or 160 µg/ml) to selected wells, the plate was transferred to an automated live cell imaging microscope (Celldiscoverer7, Zeiss) and the plate was incubated at 37 °C and 5% $CO_2$. Images were taken every 10 min over a time course of 5 h, and the average area fluorescence was determined by Image J for each time point.

## Co-incubation experiments

Fungal mycelia of *A. fumigatus* were harvested from 100 ml liquid AMM cultures that were inoculated with $1.5 \times 10^8$ conidia and grown for 2 days at 37 °C and 200 rpm. Mycelia were collected using Miracloth (Merck Millipore) and washed with 20 ml sterile ddH2O. A small amount of mycelia (~250 mg) was transferred to a new 500 ml flask containing 100 ml DMEM + 10% FCS using an inoculation loop. For expression studies of bacterial cocultures, 3 ml of a bacterial suspension in PBS at $OD_{600}$ of 7 were added after growing bacteria overnight in LB medium and washing twice with PBS. To analyse the impact of supernatants on gene expression, fungal mycelia were exposed to 25 ml of sterile-filtered supernatant collected from uninfected A549 cells or from A549 cells that were previously infected with *A. fumigatus* for 24 h. To analyse the effect of host cell lysates, 25 ml fresh DMEM + 10% FCS supplemented with 2.5 ml of ddH2O or 2.5 ml of cell lysis extract (generated from $6.3 \times 10^7$ A549 cells lysed with 5 ml ice-cold sterile ddH2O) were used. The cocultures were incubated for 4 h at 37 °C and 200 rpm, and mycelia were collected using Miracloth (Merck Millipore). The fungal mycelia were washed with 250 ml ddH2O, dried using paper towels, flash-frozen in liquid nitrogen and stored at −80 °C for subsequent RNA isolation and qRT-PCR analysis.

## CwhA-induced bacterial lysis

Bacterial overnight cultures in LB medium were diluted 1:100 in 10 ml fresh LB media and incubated for 3 h. Cells were then centrifuged at $10,000 \times g$ for 5 min and washed two times in PBS. The pellet was resuspended in 50 mM Tris buffer (pH 8) to reach an $OD_{600}$ of 1.0, and 200 µl were transferred to 1.5 ml reaction

tubes for CFU counts or to non-adherent 96-well microplates (Corning) for automated measurement of the absorbance at 600 nm using a microplate reader (TECAN infinite series). About 10 µl of CwhA diluted from a 4 mg/ml solution in PBS to final concentrations of 200–0.4 µg/ml were added, and the samples were incubated at 37 °C and 300 rpm for up to 3 h for CFU counts or without shaking at 37 °C for automated reading of OD. For CFU counts, 10 µl of the samples were collected every 15–30 min, and serial dilutions were plated on LB agar plates. PBS was used as a control.

## Origin of cell lines used in this study

A549 cells, a human type II alveolar epithelial cell line provided by American Type Culture Collection, and MH-S cells, a murine alveolar macrophage cell line, were used. Both cell lines were obtained from the American Type Culture Collection (A549: CCL-185™; MH-S: CRL-2019™), split into aliquots, and stored in liquid nitrogen. Aliquots were used to expand cells for experiments. Expanded cells were regularly tested for *Mycoplasma* infection by molecular methods.

## Infection of A549 cells for gene expression studies

For expression analysis, A549 cells were seeded in six-well microplates (TPP) at $9 \times 10^4$ cells/well in DMEM + 10% FCS and incubated for 24 h at 37 °C, 5% $CO_2$ and 21% $O_2$. Cells were washed once with PBS, and 5 ml of a spore suspension containing $9 \times 10^5$ conidia in DMEM + 10% FCS were added to the cells. The plates were incubated at 37 °C, 5% $CO_2$ and 21% $O_2$ for 9 h. For growth under hypoxia, plates were transferred to a hypoxystation (H35, Don Whitley Scientific) and incubated for an additional 6–24 h at 37 °C, 5% $CO_2$ and 0.2% $O_2$; for normoxia, plates remained at 21% $O_2$. For the 24 h incubation, the culture medium was replaced with fresh DMEM + 10% FCS to avoid starvation of cells and remove overgrowing fungi. For RNA extraction, plates were transferred on ice, cells washed with PBS, and host cells lysed with ice-cold ddH2O. Attached cells and fungi were removed from the bottom surface using a cell scraper and transferred into 1.5 ml reaction tubes. Fungal mycelia were washed once in ddH2O, collected by centrifugation at 4 °C, $13,000 \times g$ for 5 min, and the pellet was resuspended in 250 µl RL buffer (Roboklon) containing 10 µl/ml mercaptoethanol. The samples were flash-frozen in liquid nitrogen and stored at −80 °C until RNA isolation.

## Damage assays

Vegetative cells of *D. discoideum Dictyostelium discoideum* AX2 (Gerisch) (Bloomfield et al, 2008) were grown in petri dishes containing HL5 media with 1% [w/v] glucose (Formedium) at 22 °C until reaching 80% confluency. Cells were harvested by scraping and counted in the CASY cell counter (OLS Bio) and adjusted to $10^6$ cells/ml in 24-well plates with HL5 media with 1% glucose. The purified protein was added to a final concentration of 130 µg/ml, and the plate was incubated at 22 °C. Live cell numbers were determined with the CASY counter after 0, 1.5, 4, and 26 h of incubation.

To determine damage by recombinant CwhA protein, A549 cells or MH-S cells were seeded in 96-well microplates at $2 \times 10^4$ cells/well in DMEM (Gibco) or RPMI (Gibco) supplemented with 10%

FCS in a total volume of 200 µl/well. Cells were incubated for 2 days at 37 °C, 5% $CO_2$ and 21% $O_2$ to obtain a confluent layer, and the medium was exchanged for DMEM for A549 and RPMI for MH-S cells, both containing 1% penicillin /streptomycin solution (Gibco). Recombinant CwhA protein was added in a volume of 5 µl/well with final concentrations of 25, 12.5, 6.3, and 3.1 mg/ml, and the plates were incubated at 37 °C, 5%$CO_2$ and 21% $O_2$ for 24 h. About 5 µl of 5% Triton-X were added to the positive control well and incubated for another 5 min. The plates were centrifuged for 10 min at $250 \times g$ and 10 µl of the supernatant were transferred to a new microplate containing 90 µl PBS.

To determine damage during *A. fumigatus* infection, A549 cells were seeded in 24-well microplates in F-12K (Gibco) supplemented with 10% FCS in a total volume of 990 µl/well. Cells were incubated for 24 h at 37 °C, 5% $CO_2$ and 21% $O_2$ to obtain a confluent layer and the medium was exchanged for 1 ml fresh F-12K containing 1% penicillin/streptomycin solution (Gibco) containing $9 \times 10^5$ *A. fumigatus* conidia. Plates were incubated at 37 °C, 5% $CO_2$ and 21% $O_2$ for 6, 24, and 30 h. For the positive control, 50 µl of 5% Triton-X was added to the well and incubated for another 5 min. The plates were centrifuged for 10 min at 250 g and 50 µl of the supernatant were transferred to a new microplate containing 50 µl PBS.

The reaction mixture of the Roche Applied Science Cytotoxicity Detection Kit was prepared according to the manufacturer's protocol, and 100 µl of the mix was added to each well and incubated for 30 min at room temperature in the dark. The absorbance was measured using a microplate reader (TECAN Infinite series) at 492 and 690 nm for recombinant protein and a microplate reader (ClarioSTAR plus) at 490 and 680 nm for infection to detect released lactate dehydrogenase (LDH) as a read-out for cytotoxicity.

### RNA extraction and quantitative RT- PCR

Fungal mycelium was disrupted by glass beads (0.5 mm, Biospec. Products), and RNA was isolated using the GeneMatrix Universal RNA Purification Kit (Roboklon, Berlin) according to the manufacturer's instructions specified for yeast RNA isolation. RNA was eluted in 30 µl RNase-free water, the concentration was determined with Nanodrop ND-1000 (Thermo Fisher), and aliquots were prepared and stored at −80 °C until usage. RNA quality was tested using the RNA 600 Nano Kit and the Bioanalyzer 2100 (Agilent Technologies) according to the manufacturer's instructions. cDNA was generated from 800 ng of isolated RNA using Superscript III Reverse Transcriptase (Invitrogen). 2xGo Taq qPCR Mastermix (Promega) was used for PCR. Relative gene expression levels were determined by the $2^{-\Delta\Delta Ct}$ method (Schmittgen and Livak, 2008) with *cox5* and *act1* as housekeeping genes. RNA extracted from *A. fumigatus* mycelia grown in DMEM solution served as a reference.

### Determination of fungal survival in macrophages

Adherent human monocyte-derived macrophages were kindly provided by Marcel Sprenger (Leibniz Institute for Natural Product Research and Infection Biology, Jena, Germany) (Sprenger et al, 2020). Age and sex of donors was not disclosed. For infection, $5 \times 10^5$ macrophages/well were seeded in 24-well plates in RPMI supplemented with 10% FCS in a total

volume of 500 µl/well. Cells were allowed to adhere overnight at 37 °C and 5% $CO_2$ and infected with conidia of *A. fumigatus* wildtype and Δ*cwhA* at an MOI of 1 in technical triplicates for 12 and 24 h 37 °C and 5% $CO_2$. Cells were washed in PBS twice, and macrophages were lysed with 500 µl dd$H_2$0. Serial dilutions were prepared with PBS and plated onto malt agar plates. CFUs were determined after 2 days of incubation at 37 °C. For statistical analysis, the mean of the technical triplicates was used.

### PBMC isolation and stimulation

PBMCs were isolated from human blood that was drawn from healthy donors (age and sex of donors was not disclosed) into 10 ml EDTA tubes. Blood was diluted in PBS (1:1), and cell fractions were separated in lymphocyte separation medium (density 1.077 g/ml, Capricorn Scientific) by density gradient centrifugation according to the protocol supplied by the manufacturer. Cells were washed twice with PBS and resuspended in RPMI 1640 (Gibco). Cells were supplemented with 10% human serum, plated in 96-well flat-bottom plates (TPP) at a final concentration of $2.5 \times 10^6$ cells/ml and in a total volume of 200 µl. The stimulation was performed by adding $1 \times 10^7$/ml heat-killed *A. fumigatus* conidia, 10 µg/ml Muramyl-dipeptide (Invivogen), a combination of heat-killed conidia and MDP, or 25 µl of digest supernatants (details below). A combination of MDP and conidia was used as a positive (high) control, MDP only was used as a low control. After cell stimulation for 24 h, supernatants were collected and stored at −20 °C until cytokine measurement. Experiments were conducted with technical duplicates; for statistical analysis, the mean of the duplicates was used.

### Generation of digest supernatants

To prepare digest supernatants, the following combinations were incubated for 16 h at 37 °C: (i) 50 µl peptidoglycan derived from *S. aureus* (*S. aureus* PG), (ii) 50 µl *S. aureus* PG + 20 µg CwhA, (iii) 20 µg CwhA. The reaction was stopped by boiling at 98 °C for 5 min, and non-soluble *S. aureus* PG was removed by centrifugation and transfer of the supernatants to fresh tubes.

### Cytokine measurements

The cytokine levels in stimulated PBMC supernatants were quantified using the uncoated ELISA kits for IL-1β and TNF-α cytokines (Invitrogen) according to the protocols supplied by the manufacturer.

### Statistical analysis

Statistical analyses were performed with GraphPad version 9.12. Details on the test used for each data set and the number of replicates are stated in the respective figure/table legends.

## Data availability

The data supporting the findings of this study are available as Source Data. The UPLC/MS spectra used to analyse the muropeptide profiles are available in the BioStudies database under accession number S-BSST1929.

The source data of this paper are collected in the following database record: biostudies:S-SCDT-10_1038-S44319-025-00508-3.

## Peer review information

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

## Acknowledgements

We thank Maria Stroe, Carmen Schult, and Dirk Femerling for excellent technical assistance, Ralf Ehricht and Stefan Monecke for providing MRSA strains, Sven Krappmann for the plasmid pSK379 and advice on the overexpression strategy, and Mark Gresnigt for encouraging discussions. The work was financially supported by the BMBF (Forschungscampus InfectoGnostics, project IDES, FKZ: 13GW0096E, to IDJ) and the German Research Foundation (DFG; TRR 124 FungiNet, "Pathogenic fungi and their human host: Networks of Interaction," DFG project number 210879364, Project Z2 to FvE). The synopsis figure was created using BioRender.com.

## Author contributions

**Silke Machata**: Conceptualization; Resources; Data curation; Formal analysis; Validation; Investigation; Visualization; Methodology; Writing—original draft; Writing—review and editing. **Ute Bertsche**: Data curation; Formal analysis; Investigation; Visualization; Methodology; Writing—review and editing. **Franziska Hoffmann**: Formal analysis; Investigation; Visualization; Writing—review and editing. **Zaher M Fattal**: Investigation. **Franziska Kage**: Formal analysis; Investigation; Visualization; Writing—review and editing. **Michal Flak**: Investigation. **Alexander N J Iliou**: Investigation. **Falk Hillmann**: Formal analysis; Supervision; Investigation; Writing—review and editing. **Ferdinand von Eggeling**: Formal analysis; Supervision; Investigation; Visualization; Writing—review and editing. **Hortense Slevogt**: Methodology; Writing—review and editing. **Axel A Brakhage**: Supervision; Writing—review and editing. **Ilse D Jacobsen**: Conceptualization; Formal analysis; Supervision; Funding acquisition; Validation; Visualization; Writing—original draft; Project administration; Writing—review and editing.

Source data underlying figure panels in this paper may have individual authorship assigned. Where available, figure panel/source data authorship is listed in the following database record: biostudies:S-SCDT-10_1038-S44319-025-00508-3.

## Disclosure and competing interests statement

The authors declare no competing interests.

# Expanded View Figures

**Figure EV1.  Effects of CwhA on bacterial growth in LB media and Tris buffer.**

(**A**) Bacterial growth in LB media supplemented with CwhA. The optical density at 600 nm was measured every 30 min in a plate reader after the addition of PBS or different concentrations of recombinant CwhA (25–200 μg/ml). One representative experiment is shown. (**B**) The optical density (600 nm) after addition of recombinant CwhA (16–200 μg/ml) to bacterial suspensions in 50 mM Tris buffer (pH 8) was measured overtime. Data from a single experiment is shown. Data from a biological replicate is shown in Fig. 3A. Source data are available online for this figure.

▶

## A

## Effect of CwhA in LB

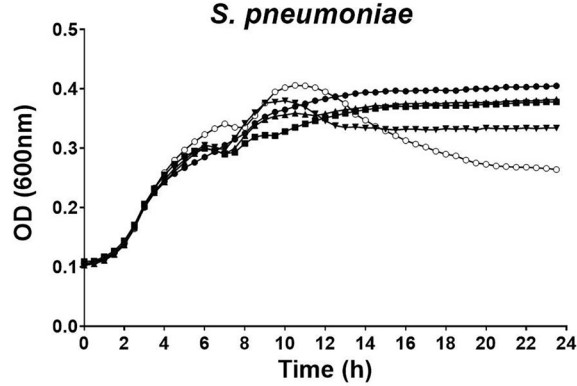

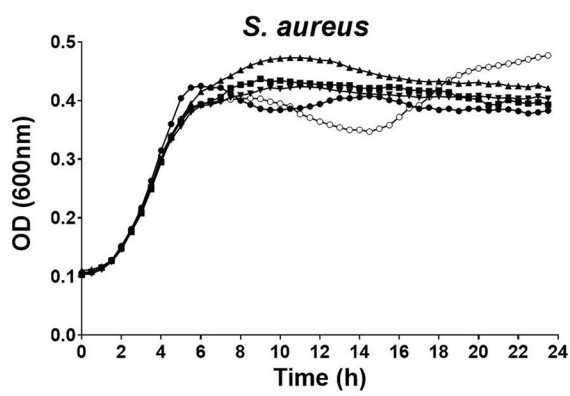

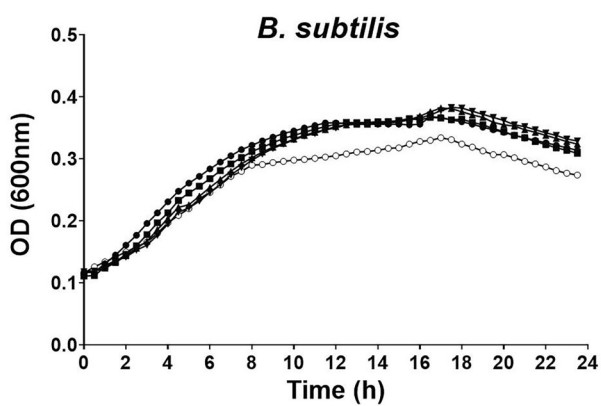

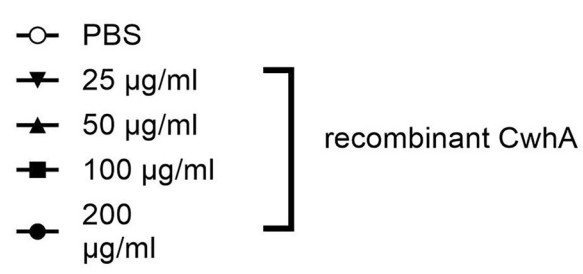

## B

# Dose-dependent effect of CwhA on *S. aureus* Newman in Tris buffer

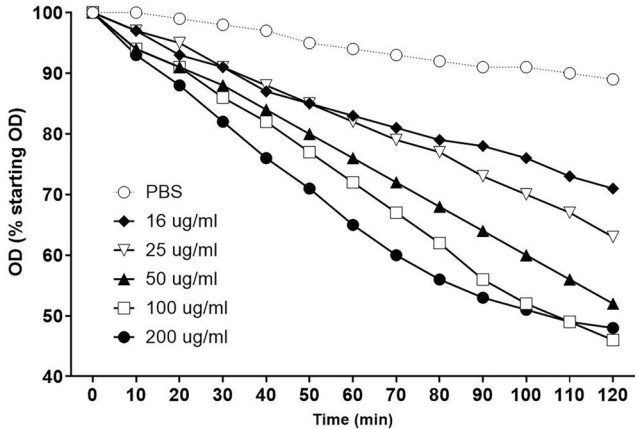

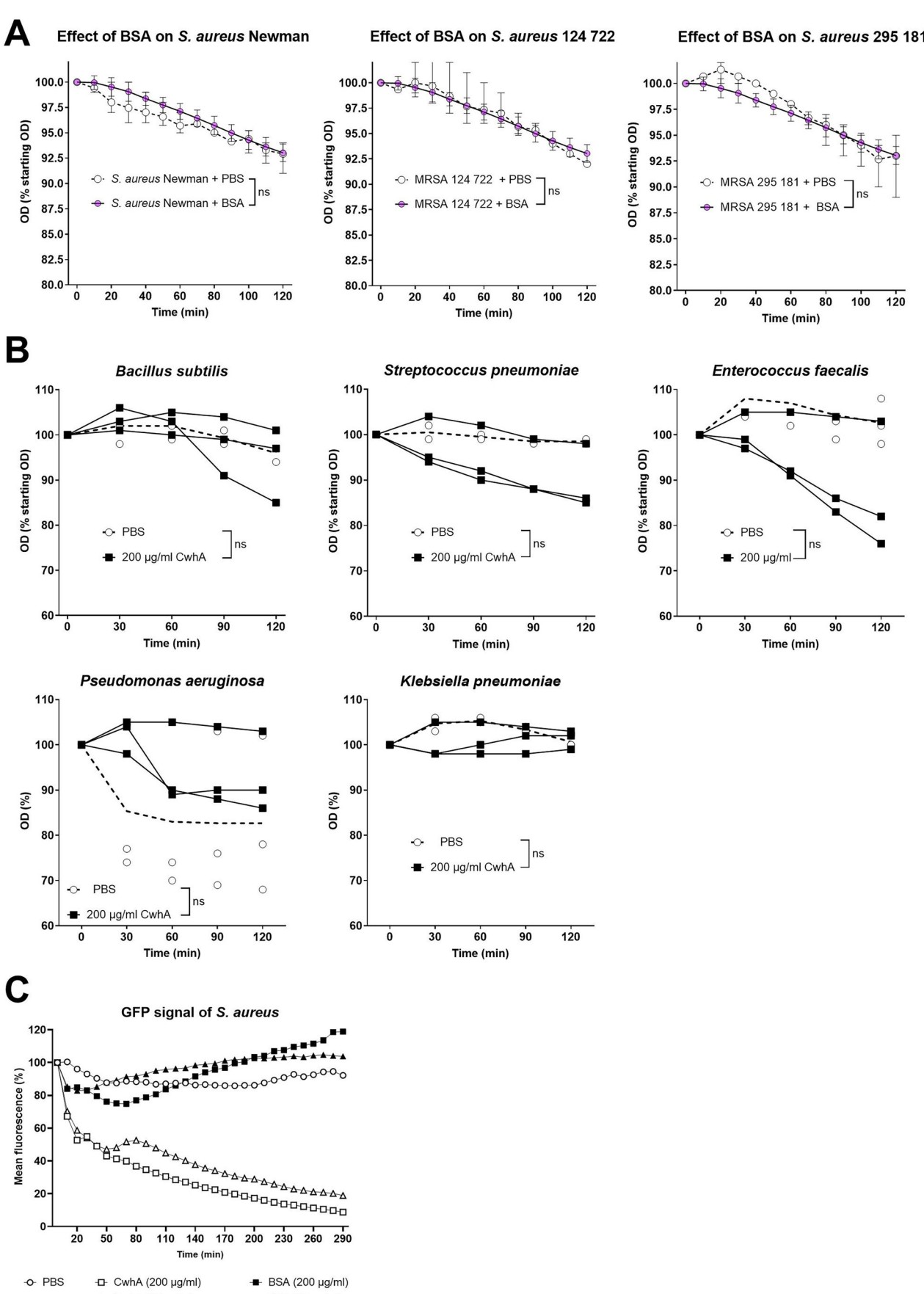

**Figure EV2. Effect of BSA on *S. aureus* and CwhA-mediated lysis of different bacterial species.**

(A, B) The optical density (OD) at 600 nm after addition of PBS, BSA (200 mg/ml) or recombinant CwhA (200 mg/ml) to bacterial suspensions ($n = 3$ biological replicates) in 50 mM Tris buffer (pH 8) was measured at the indicated time points. (A) Three different *S. aureus* strains in Tris buffer after addition of either PBS or BSA. The PBS data is also shown in Fig. 3B. (B) OD of different bacterial species after the addition of either PBS or CwhA. (C) GFP signal of a GFP-expressing *S. aureus* strain after treatment with CwhA. Bovine serum albumin (BSA) and PBS were used as controls. One of $n = 2$ biological replicates, see also Fig. 3D. Data information: In (A, B) data were shown as individual data points and the mean as connecting line. ns no statistically significant difference between the groups (two-way ANOVA). Source data are available online for this figure.

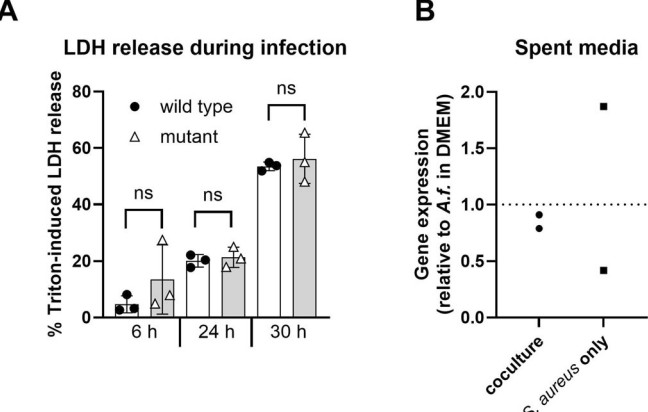

**Figure EV3. Host cell damage during *A. fumigatus* infection, and the effect of spent media on *cwhA* expression.**

(A) A549 cells were seeded in 24-well microplates and infected with $9 \times 10^5$ *A. fumigatus* conidia. Plates were incubated at 37 °C, 5% $CO_2$ and 21% $O_2$ for either 6, 24, or 30 h. LDH released into the supernatant of infected cells was normalized to the Triton-lysed positive control. $n = 3$ biological replicates per treatment and time point. (B) *A. fumigatus* was grown in spent media (sterile-filtered culture supernatant) of *S. aureus* Newman cultured alone or in coculture with *A. fumigatus* for 4 h in DMEM. $n = 2$ biological replicates. Data information: Data in (A) are shown as a scatter plot with bars (mean) and SD (error bars). ns not significant (two-tailed unpaired *t*-test). In (B), data were represented as individual data points. Source data are available online for this figure.

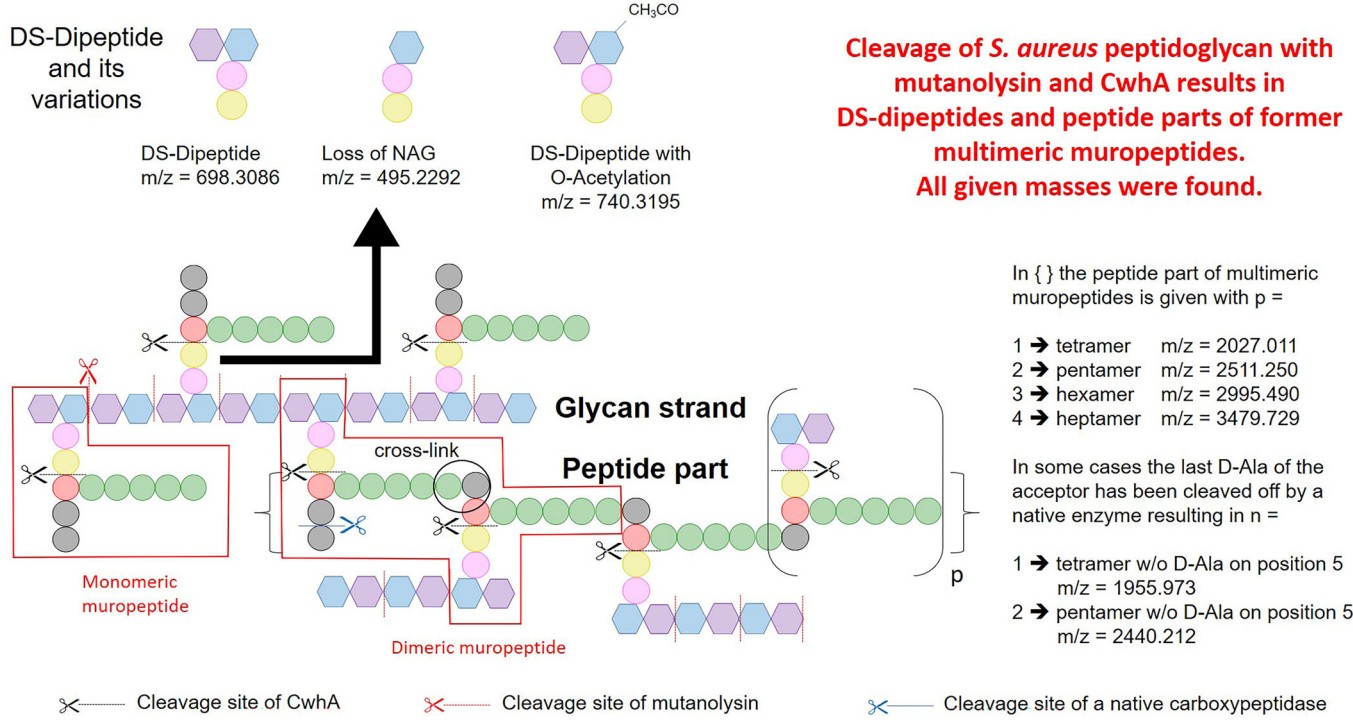

**Figure EV4.**   Schematic depiction of peptidoglycan cleavage by mutanolysin and CwhA.

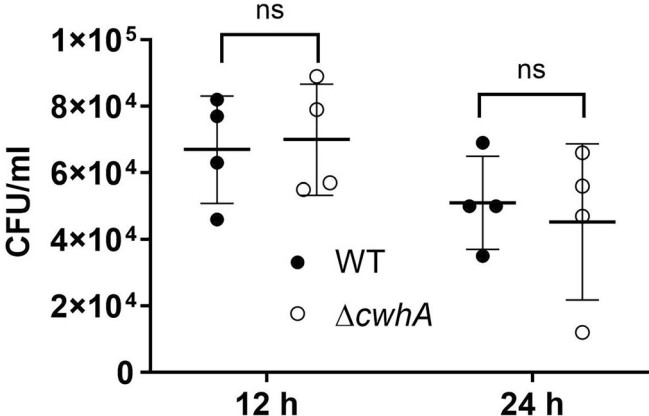

**Figure EV5.  Survival of *Aspergillus fumigatus* wildtype (WT) and Δ*cwhA* in human macrophages.**

Differentiated adherent human monocyte-derived macrophages ($n = 4$ different donors - biological replicates) were infected with fungal conidia at an MOI of 1 for 12 and 24 h and lysed to determine fungal growth on malt agar plates. Data information: Data from four independent donors are shown as a scatter dot plot with mean ± SD. ns not significant (one-way ANOVA and Tukey's multiple comparisons).  Source data are available online for this figure.

