## [Peer Review File · EMBO Reports]

Identification of a fungal antibacterial endopeptidase that cleaves peptidoglycan

Silke Machata, Ute Bertsche, Franziska Hoffmann, Zaher Fattal, Franziska Kage, Michal Flak, Alexander Iliou, Falk Hillmann, Ferdinand von Eggeling, Hortense Slevogt, Axel Brakhage, and Ilse Jacobsen

Corresponding author(s): Ilse Jacobsen (ilse.jacobsen@leibniz-hki.de)

Review Timeline:

Submission Date:	11th Sep 24
Editorial Decision:	7th Oct 24
Revision Received:	21st Mar 25
Editorial Decision:	30th Apr 25
Revision Received:	6th Jun 25
Accepted:	12th Jun 25

Editor: Achim Breiling

Transaction Report:

Dear Dr. Jacobsen,

Thank you for the submission of your manuscript to EMBO reports. I have now received the reports from the three referees that were asked to evaluate your study, which can be found at the end of this email.

As you will see, the referees find the study very interesting. Nevertheless, they have several comments, concerns, and suggestions, indicating that a major revision of the manuscript is necessary to allow publication of the study in EMBO reports. As the reports are below, and all the concerns need to be addressed, I will not detail them further here.

Given the constructive referee comments, I would like to invite you to revise your manuscript with the understanding that the concerns of the referees must be addressed in the revised manuscript and in a detailed point-by-point response. Acceptance of your manuscript will depend on a positive outcome of a second round of review. It is EMBO reports policy to allow a single round of revision only and acceptance of the manuscript will therefore depend on the completeness of your responses included in the next, final version of the manuscript.

- 1) a .docx formatted version of the final manuscript text (including legends for main figures, EV figures and tables), but without the figures included. Figure legends should be compiled at the end of the manuscript text.
- 2) individual production quality figure files as .eps, .tif, .jpg (one file per figure), of main figures and EV figures. Please upload these as separate, individual files upon re-submission.

- 4) a complete author checklist, which you can download from our author guidelines

(<https://www.embopress.org/page/journal/14693178/authorguide>). Please insert page numbers in the checklist to indicate where the requested information can be found in the manuscript. The completed author checklist will also be part of the RPF.

- 5) that primary datasets produced in this study (e.g. RNA-seq, ChIP-seq, structural and array data) are deposited in an

appropriate public database. If no primary datasets have been deposited, please also state this in a dedicated section (e.g. 'No primary datasets have been generated and deposited'), see below.

The accession numbers and database should be listed in a formal "Data Availability" section that follows the model below. This is now mandatory (like the COI statement). Please note that the Data Availability Section is restricted to new primary data that are part of this study. This section is mandatory. As indicated above, if no primary datasets have been deposited, please state this in this section

Data availability

8) Regarding data quantification and statistics, please make sure that the number "n" for how many independent experiments were performed, their nature (biological versus technical replicates), the bars and error bars (e.g. SEM, SD) and the test used to calculate p-values is indicated in the respective figure legends (also for EV and Appendix figures). Please also check that all the p-values are explained in the legend, and that these fit to those shown in the figure. Please provide statistical testing where applicable. Please avoid the phrase 'independent experiment', but clearly state if these were biological or technical replicates. Please also indicate (e.g. with n.s.) if testing was performed, but the differences are not significant. In case n=2, please show the data as separate datapoints without error bars and statistics. See also: <http://www.embopress.org/page/journal/14693178/authorguide#statisticalanalysis>

9) Please add scale bars of similar style and thickness to microscopic images, using clearly visible black or white bars (depending on the background). Please place these in the lower right corner of the images themselves. Please do not write on or near the bars in the image but define the size in the respective figure legend.

10) Please also note our reference format:

12) We now use CRediT to specify the contributions of each author in the journal submission system. CRediT replaces the author contribution section. Please use the free text box to provide more detailed descriptions and do NOT provide your final manuscript text file with an author contributions section. See also our guide to authors: <https://www.embopress.org/page/journal/14693178/authorguide#authorshipguidelines>

13) All Materials and Methods need to be described in the main text using our 'Structured Methods' format, which is required for

all research articles. According to this format, the Methods section should include a Reagents and Tools Table (listing key reagents, experimental models, software, and relevant equipment and including their sources and relevant identifiers), uploaded as separate file, followed by a Methods section in which we encourage the authors to describe their methods using a step-by-step protocol format with bullet points, to facilitate the adoption of the methodologies across labs. More information on how to adhere to this format as well as downloadable templates (.doc) for the Reagents and Tools Table can be found in our author guidelines (section 'Structured Methods'):

14) Please add up to 5 keywords to the manuscript and order the manuscript sections like this, using these names: Title page - Abstract - Keywords - Introduction - Results - Discussion - Methods - Data availability section - Acknowledgements (including funding information) - Disclosure and Competing Interests Statement - References - Figure legends - Expanded View Figure legends

15) Please make sure that all the funding information is also entered into the online submission system and that it is complete and similar to the one in the acknowledgement section of the manuscript text file.

I look forward to seeing a revised form of your manuscript when it is ready.

Yours sincerely,

Referee #1:

The presented study by investigates a secreted *A. fumigatus* protein with antibacterial properties. The team determined endopeptidase activity and named the protein *cwhA*. The enzyme cleaves peptidoglycan of Gram-positive bacteria at specific residues within the peptidoglycan stem peptide. *CwhA* is abundant during experimental aspergillosis and expression can be induced by live bacteria.

The manuscript is very well written and data are presented adequately and in a compelling manner. The discovery of the first fungal wall-lysing antibacterial endopeptidase is of general importance to understand virulence mechanisms of *A. fumigatus*. The presented finding could inspire novel bactericidal medication against multi-resistant Gram-positive pathogens.

Major comments:

The data presentation is exceptionally clear. I only have one issue. In Figure 5A, the authors attempt to demonstrate under which conditions *cwhA* is expressed and they use epithelial cells as potential stimulant for *cwhA* expression. They speculate that "Increased expression of *cwhA* could only be observed at a late stage of infection, which may be attributable to a response to factors released from damaged host cells, or be merely the result of nutritional starvation."

A) The authors could test whether *cwhA* expression coincides with the damage / cell death of cocultured epithelial cells to make their point clearer.

B) In Fig. 5B they show that exposure to bacteria is a sufficient inducing stimulus for *cwhA* expression. At first glance both findings seem a bit counterintuitive which should be clarified. In light of findings which show that bacterial supernatants are not sufficient to induce *cwhA* expression, it seems plausible that PAMPs and DAMPs could be a responsible trigger. In the case of PAMPs they might not be released by bacteria and hence cannot be found in the culture supernatant. The authors could test

whether contrarily, filtered supernatants from damaged host cells would be sufficient to trigger *cwhA* expression.

C) Furthermore, later in the text the authors exclude mere starvation of *A. fumigatus* as an inducing condition for *cwhA* contradicting their speculation cited above. Since the authors do not offer any data to back the speculation and later themselves allude to the speculations disproof they should remove the speculation entirely.

Minor comments:

-Figure 1, figure legend

In the legend, authors should state of how many repetitions the representative images were chosen.

-Figure 3, figure legend

The authors mentioned that several means stem from two repetitions only. In that case, the error bars should be removed at those particular graphs or time points.

- Figure legends in general, when data from only two independent experiments were used to generate a mean value, no statistical test or error bars can be applied.

-Materials and Methods

LC-MS analysis of cleavage products

In the sentence "from Waters with a guard column on an Agilent 1290 and a G42220 Bin Pump, baryp running in UHPLC mode with a maximum pressure limit of 1200 bar" there seems to be an error

Referee #2:

The authors present an interesting and relevant study that addresses an antimicrobial activity secreted by *Aspergillus fumigatus*, an opportunistic fungal pathogen that commonly thrives in decaying organic material where it is part of a competitive microbiome. In order to sustain in such a challenging ecological niche, fungi have evolved numerous natural products that serve as bioactive compounds to support competitiveness. Secreted peptides or proteins with antimicrobial activities of fungal origin have only been characterized to a limited extent, and the study by Machata et al. adds valuable insights in this respect, albeit falling short in elucidating the role of *CwhA* in the corresponding context, that is inter-microbial competition and colonisation resistance. Especially the latter aspect appears to be of utmost relevance when studying an antibacterial activity of an opportunistic fungal pathogen; yet, shedding light on this may be beyond the scope of this piece of work that takes first and fundamental steps in characterising the *cwhA* gene and its product on several levels.

Besides this general remark, there are several issues that might be addressed in a revised version of the manuscript, which are as follows:

The Title is exaggerating and actually misleading, as immune modulation is an indirect effect resulting from *CwhA* chewing up bacterial peptidoglycans: It is the breakdown products that might be sensed by immune cells to result in altered cytokine levels and not the endopeptidase per se to have an effect. This aspect is more plausibly addressed in the Abstract, the final half-sentence of which reads somehow incorrect.

When elaborating on secondary metabolites of *A. fumigatus* being relevant for infection, a more comprehensive review on the topic might be cited as well (and not only two compounds that are at the center of attention for the authors). *Afusins* of *A. fumigatus* are secreted during conidiation and their activity might actually be restricted against Gram-positives. Albeit the initial identification of *CwhA* as BOYAYO had been described before, it might be worth spending some more pieces of background information before listing the novel insights of this study.

In the Results part, most of the experimental approaches and corresponding lines of reasoning appear as valid and reasonable. Yet, a deeper analysis of transcriptomic data collections could have been performed to maybe gain a clue about the environmental factors that trigger *CwhA* expression (as the presented pieces of data are rather puzzling). The extensive analyses on substrate specificity and cleavage patterns contribute to the study's validity, while the section on immunomodulation reads rather ambiguous by postulating in vivo effects based on NOD2 but then remaining at the in vitro level without addressing the presumed role of this intracellular PRR. The Discussion presents several interesting and relevant lines of reasoning but omits aspects related to the pulmonary microbiome, inter-microbial competition, and, consequently, colonisation resistance. Considering the mentioned defensins of fungal origin, the authors might revise the final statement that "*CwhA* is the first secreted antibacterial protein identified in *A. fumigatus*."

The Methods section reads detailed and comprehensive with only minor ambiguities, such as the use of recombinant *CwhA* for peptide mass correlation in the process of IMS detection of this very protein, amplification of the *CwhA* coding sequence that is interrupted by one intron from genomic DNA, or the description of PBMC stimulation that is interwoven with the protocol for

digest supernatants preparation.

The reference of Kale et al. (2017) Sci. Rep. that had been cited in the preliminary study appears to be missing here.

In essence, a comprehensive multi-level characterisation of an anti-microbial activity secreted by an opportunistic fungal pathogen is presented that opens new avenues for further investigation of inter-microbial competition and fungal virulence and that definitively merits publication.

Referee #3:

This manuscript from Jacobsen and colleagues identifies a novel, secreted fungal endopeptidase (CwhA) that is capable of cleaving bacterial peptidoglycan (PG). The authors nicely characterised that CwhA can cleave PG ultimately being able to lead to bacterial lysis. Moreover, they provide robust proteomic data to support its expression in the IPA lung near the fungal lesion. However, other claims are only weakly supported by the provided data. Moreover, there are a number of typographical errors and missing or mislabel figure panels.

- 1) Figure panels 3B and 3C are mislabeled in the text.
- 2) In Figure 3, BSA should be used as a non-specific protein control in panels A, B, and C as well.
- 3) Figure 4 - Why is there no decrease of the cwhA-null mutant? Does a phenotype get pulled out in a competitive assay? These data suggest CwhA is sufficient for the bacterial killing but might not be necessary.
- 4) The text says 20% O₂ for normoxia but the figure says 21%, please make sure they are congruent with one another.
- 5) Figure 5 - Host cell data, which is claimed in the text to be 5B, is not found in the manuscript.
- 6) Figure 6 - These data are overstated. For Figure 6A, how can the authors say there was no damage to amoeba when at 26h the cell counts are down? Also, are cell counts the best measure of amoeba health? Figure 6B only goes to a 25 ug/ml dose on mammalian cells but up to 200 ug/ml was used in the bacterial killing assays in Figure 3. From Figure 6C the authors state that CwhA had a stimulatory impact on fungal growth but there are no statistics to base that statement on.
- 7) Figure 8 - The figure legend lacks what statistical test was used on the data. Also, the text says IL-1alpha was measured but the figure says IL-1beta, please make sure they are congruent with one another
- 8) Figure 8 - PG is also a TLR2 agonist, why was PG alone not sufficient drive inflammatory cytokine expression. Also, they authors need to demonstrate that NOD2 is essential for sensing the PG treated with CwhA, there is no evidence to support the claim that NOD2 is involved.

We thank the reviewers and Editor for the thorough reading of our manuscript and the constructive comments. We have carefully addressed all points as follows:

Referee #1:

The presented study by investigates a secreted *A. fumigatus* protein with antibacterial properties. The team determined endopeptidase activity and named the protein *cwhA*. The enzyme cleaves peptidoglycan of Gram-positive bacteria at specific residues within the peptidoglycan stem peptide. *CwhA* is abundant during experimental aspergillosis and expression can be induced by live bacteria.

The manuscript is very well written and data are presented adequately and in a compelling manner. The discovery of the first fungal wall-lysing antibacterial endopeptidase is of general importance to understand virulence mechanisms of *A. fumigatus*. The presented finding could inspire novel bactericidal medication against multi-resistant Gram-positive pathogens.

Major comments:

The data presentation is exceptionally clear. I only have one issue. In Figure 5A, the authors attempt to demonstrate under which conditions *cwhA* is expressed and they use epithelial cells as potential stimulant for *cwhA* expression. They speculate that "Increased expression of *cwhA* could only be observed at a late stage of infection, which may be attributable to a response to factors released from damaged host cells, or be merely the result of nutritional starvation."

A) The authors could test whether *cwhA* expression coincides with the damage / cell death of cocultured epithelial cells to make their point clearer.

We quantified LDH release of infected cells over time, and observed a clear increase at late time points. This data is included in the revised manuscript as Supplementary Figure 3.

B) In Fig. 5B they show that exposure to bacteria is a sufficient inducing stimulus for *cwhA* expression. At first glance both findings seem a bit counterintuitive which should be clarified. In light of findings which show that

bacterial supernatants are not sufficient to induce *cwhA* expression, it seems plausible that PAMPs and DAMPs could be a responsible trigger. In the case of PAMPs they might not be released by bacteria and hence cannot be found in the culture supernatant. The authors could test whether contrarily, filtered supernatants from damaged host cells would be sufficient to trigger *cwhA* expression.

We thank the reviewer for this suggestion and have analyzed *cwhA* expression of mycelia exposed to filtered supernatants of uninfected A549 cells, supernatants of A549 cells infected with *A. fumigatus*, cell culture media supplemented with lysed A549 cells or H₂O (H₂O control), or supernatants of uninfected A549 cells with lysed A549 cells (new Fig. 5B in the revised manuscript). Only supernatant from infected cells led to an increase in *cwhA* expression. This excludes DAMPs that are present within intact cells (constitutive DAMPs, eg DNA) as the trigger for upregulation of *cwhA*. It is, however, possible that other DAMPs that are upregulated in or expressed only by damaged cells (inducible DAMPs) play a role. We have included this in the revised discussion.

C) Furthermore, later in the text the authors exclude mere starvation of *A. fumigatus* as an inducing condition for *cwhA* contradicting their speculation cited above. Since the authors do not offer any data to back the speculation and later themselves allude to the speculations disapproval they should remove the speculation entirely.

Based on the new experiments with supernatants and host cell lysates, we think that it is unlikely that nutritional starvation is a relevant signal for *cwhA* expression. We have removed the respective parts from the manuscript.

Minor comments:

-Figure 1, figure legend

In the legend, authors should state of how many repetitions the representative images were chosen.

We analyzed two sections each of five lungs. This information was added to the figure legend.

-Figure 3, figure legend

The authors mentioned that several means stem from two repetitions only. In that case, the error bars should be removed at those particular graphs or time points.

We fully agree that it would not be appropriate to indicate SEM or SD for experiments that were only repeated twice. However, as stated in the figure legend, in the relevant graphs in Figure 3 (A, D), which are based on two experiments each, the "error" bars indicated the range. For more clarity, we decided to show only one out of the two experiments in Figure 3A and D, respectively, in the revised manuscript and included the data of the second experiment (demonstrating reproducibility) as Supplementary Figure 1B and 2C.

- Figure legends in general, when data from only two independent experiments were used to generate a mean value, no statistical test or error bars can be applied.

We fully agree. As mentioned above, in graphs depicting results from only two experiments, we used bars to indicate the range, and stated that clearly in the figure legend. As this might be overlooked, we agree that for improved clarity it is better to remove the "error" bars, and have done this for all affected figures (Figure 3A, D; 5D; 6B, D). No statistical tests were applied to these data sets in the original manuscript.

-Materials and Methods

LC-MS analysis of cleavage products

In the sentence "from Waters with a guard column on an Agilent 1290 and a G42220 Bin Pump, baryp running in UHPLC mode with a maximum pressure limit of 1200 bar" there seems to be an error

Thank you for pointing out this mistake. We have corrected the sentence as follows. "Muropeptide separation was performed with a Waters ACQUITY UHPLC column (CSH C18 1.7 μm , 2.1x150 mm) on an Agilent 1290 and a G42220 Binary Pump running in UHPLC mode with a maximum pressure limit of 1200 bar."

Referee #2:

The authors present an interesting and relevant study that addresses an antimicrobial activity secreted by *Aspergillus fumigatus*, an opportunistic fungal pathogen that commonly thrives in decaying organic material where it is part of a competitive microbiome. In order to sustain in such a challenging ecological niche, fungi have evolved numerous natural products that serve as bioactive compounds to support competitiveness. Secreted peptides or proteins with antimicrobial activities of fungal origin have only been characterized to a limited extent, and the study by Machata et al. adds valuable insights in this respect, albeit falling short in elucidating the role of CwhA in the corresponding context, that is inter-microbial competition and colonisation resistance. Especially the latter aspect appears to be of utmost relevance when studying an antibacterial activity of an opportunistic fungal pathogen; yet, shedding light on this may be beyond the scope of this piece of work that takes first and fundamental steps in characterising the cwhA gene and its product on several levels. Besides this general remark,

there are several issues that might be addressed in a revised version of the manuscript, which are as follows:

The Title is exaggerating and actually misleading, as immune modulation is an indirect effect resulting from CwhA chewing up bacterial peptidoglycans: It is the breakdown products that might be sensed by immune cells to result in altered cytokine levels and not the endopeptidase per se to have an effect. This aspect is more plausibly addressed in the Abstract, the final half-sentence of which reads somehow incorrect.

We have changed the title to the following: "Identification of a fungal antibacterial endopeptidase that cleaves peptidoglycan".

When elaborating on secondary metabolites of *A. fumigatus* being relevant for infection, a more comprehensive review on the topic might be cited as well (and not only two compounds that are at the center of attention for the authors). Afusins of *A. fumigatus* are secreted during conidiation and their activity might actually be restricted against Gram-positives.

We have changed the relevant part in the introduction as follows to highlight that several secondary metabolites with antibacterial activities have been identified, and have included additional references: "Conversely, a recent microbiome analysis suggests that *A. fumigatus* colonization shapes the lung microbiome (Mirhakkak *et al*, 2023). This likely involves *A. fumigatus* secondary metabolites, several of which have been shown to be antibacterial and whose production is affected by interactions with bacteria (Boysen *et al*, 2021; Krespach *et al*, 2023; Margalit *et al*, 2022; Netzker *et al.*, 2018; Stroe *et al*, 2020). One example for a secondary metabolite with antibacterial activity is gliotoxin, which inhibits *P. aeruginosa* in co-culture experiments (Reece *et al*, 2018). Another is fumigermin that inhibits germination of bacterial spores (Stroe *et al.*, 2020). Moreover, defensin-like peptides called afusins are produced during *A. fumigatus* conidiation and mediate protection of conidia against bacteria (Dümig *et al*, 2021)."

Albeit the initial identification of CwhA as B0YAY0 had been described before, it might be worth spending some more pieces of background information before listing the novel insights of this study.

We included more information on why we decided to further study B0YAY0 in the beginning of the results section. We specifically highlight that the main rationale for further studying B0YAY0 was its high expression *in vivo* and the possibility that it would function as a bacterial cell wall hydrolase.

In the Results part, most of the experimental approaches and corresponding lines of reasoning appear as valid and reasonable. Yet, a deeper analysis of transcriptomic data collections could have been performed to maybe gain a clue about the environmental factors that trigger CwhA expression (as the presented pieces of data are rather puzzling).

We have added all information on B0YAY0 expression we could find by searching published transcriptomic data for a mention of the respective gene to the beginning of the revised Results section.

The extensive analyses on substrate specificity and cleavage patterns contribute to the study's validity, while the section on immunomodulation reads rather ambiguous by postulating *in vivo* effects based on NOD2 but then remaining at the *in vitro* level without addressing the presumed role of this intracellular PRR.

Unfortunately, we could not obtain NOD2 ko mice within a reasonable time frame (the line is currently not actively bred at Jackson Laboratories, and we found no active breeding colony in our state), and thus, could not experimentally test the role of CwhA-mediated PG digestion and NOD2 for the host response *in vivo*. While we'll try to follow up on this in the future, this seems out of scope for this manuscript, and, consequently, we have changed the manuscript text to reflect that sensing by NOD2 is speculative and requires further investigations.

The Discussion presents several interesting and relevant lines of reasoning but omits aspects related to the pulmonary microbiome, inter-microbial competition, and, consequently, colonisation resistance.

We thank the reviewer for this suggestion and have extended the relevant part of the discussion.

Considering the mentioned defensins of fungal origin, the authors might revise the final statement that "CwhA is the first secreted antibacterial protein identified in *A. fumigatus*."

To our knowledge, CwhA is the first actual protein with antibacterial properties produced by *A. fumigatus*, but it is obviously not the only antibacterial factor. To clarify this, we have extended the sentence as follows: "In summary, CwhA is the first secreted antibacterial protein identified in *A. fumigatus*, complementing other fungal antibacterial factors such as secondary metabolites and defensins."

The Methods section reads detailed and comprehensive with only minor ambiguities, such as the use of recombinant CwhA for peptide mass correlation in the process of IMS detection of this very protein, amplification of the CwhA coding sequence that is interrupted by one intron from genomic DNA, or the description of PBMC stimulation that is interwoven with the protocol for digest supernatants preparation.

We have revised the methods section as follows:

1. The original sentence was potentially misleading; the recombinant protein served as a control for peptides being generated by the tryptic digest (in addition to in silico predicted peptides). The sentence was changed accordingly.
2. The primer used for gene amplification were designed to amplify the gene excluding the signal peptide, and thus binds after the intron. We have added this information to the Methods section for clarity.
3. The protocol for digest supernatants preparation was moved to a separate paragraph following the protocol on PBMC stimulation.

The reference of Kale et al. (2017) Sci. Rep. that had been cited in the preliminary study appears to be missing here.

We reference Kale et al. at the revised beginning of the results part in the first paragraph, in which we summarize what is known about CwhA based on previous publications.

Referee #3:

This manuscript from Jacobsen and colleagues identifies a novel, secreted fungal endopeptidase (CwhA) that is capable of cleaving bacterial peptidoglycan (PG). The authors nicely characterised that CwhA can cleave PG ultimately being able to lead to bacterial lysis. Moreover, they provide robust proteomic data to support its expression in the IPA lung near the fungal lesion. However, other claims are only weakly supported by the provided data. Moreover, there are a number of typographical errors and missing or mislabel figure panels.

1) Figure panels 3B and 3C are mislabeled in the text.

The mislabeling occurred in the figure, we have corrected this.

2) In Figure 3, BSA should be used as a non-specific protein control in panels A, B, and C as well.

The thank the reviewer for this suggestion. We have measured the effect of BSA (200 µg/ml) on *S. aureus* OD in Tris buffer for three different strains and found no difference to the PBS control. The new data is presented as Supplementary Figure 2A.

3) Figure 4 - Why is there no decrease of the *cwhA*-null mutant? Does a phenotype get pulled out in a competitive assay? These data suggest CwhA is sufficient for the bacterial killing but might not be necessary.

As shown in Fig. 4B, the wildtype does not produce detectable amounts of CwhA under the culture conditions used. This is consistent with gene expression data showing that CwhA is not expressed *in vitro* – contrary to the *in vivo* condition. Consistent with this, co-incubation of *S. aureus* with *A. fumigatus* wildtype did not lead to a decrease in *S. aureus* CFU (Figure 4C). In the absence of production by the wildtype, one would not expect the gene deletion to have an effect.

4) The text says 20% O₂ for normoxia but the figure says 21%, please make sure they are congruent with one another.

We have changed the text to 21% O₂ to make this consistent with the figure.

5) Figure 5 - Host cell data, which is claimed in the text to be 5B, is not found in the manuscript.

The text should refer to Figure 5A, which shows expression data during infection of host cells. The text has been changed accordingly. Please note that the subsequent parts of Figure 5 where likewise mis-referred, and that this has also been corrected.

6) Figure 6 - These data are overstated. For Figure 6A, how can the authors say there was no damage to amoeba when at 26h the cell counts are down? Also, are cell counts the best measure of amoeba health?

Figure 6B only goes to a 25 µg/ml dose on mammalian cells but up to 200 µg/ml was used in the bacterial killing assays in Figure 3. From Figure 6C the authors state that CwhA had a stimulatory impact on fungal growth but there are no statistics to base that statement on.

Amoeba:

A lower number of amoeba was indeed observed after 26 h in the presence of 130 µg/ml CwhA compared to the control – as clearly stated in the original manuscript. It should, however, be noted that amoeba cell numbers increased over 10fold from 1.5 h to 26 h in the presence of CwhA. While cell counts do not necessarily reflect the number of viable cells, a 10fold increase in cell numbers can only be explained by active proliferation. Thus, there was substantial growth of amoeba in the presence of CwhA, which argues against an overt toxic effect. We have revised this section of the results to clarify these findings and our interpretation. Of note, for *S. aureus* substantial killing was observed within 2 h with 100 µg/ml and less, based not only on CFU but also OD (Figure 1).

Mammalian cells:

For technical reasons, we were limited in the volume that could be applied to the mammalian cells, and thus were not able to test higher doses of CwhA. However, we'd like to point out that doses as low as 16 µg/ml had a clear effect on *S. aureus* within less than 2 h (Fig. 1A, D), whereas no effect was observed for mammalian cells treated for 24 h with 25 µg/ml CwhA.

A. fumigatus planktonic growth (Fig. 6C):

Quantification of *A. fumigatus* biomass in planktonic cultures is not trivial, since OD measurements and CFU counts are not reliable for mycelial growth. Theoretically, one could determine the dry weight of cultures. However, we think that the visual impression of cultures depicted in Figure 6C clearly demonstrates better growth of *A. fumigatus* in the presence of

CwhA compared to the PBS control. In addition, quantification of biofilm formation (Fig. 6D) supports the notion that CwhA has no negative impact on fungal growth. We do agree that comparison to BSA is difficult and have re-written the text accordingly.

To avoid overstating any of the results, we have changed the paragraph heading of this results part to "Recombinant CwhA causes no overt damage to amoeba, host cells, and fungal cells".

7) Figure 8 - The figure legend lacks what statistical test was used on the data

Also, the text says IL-1alpha was measured but the figure says IL-1beta, please make sure they are congruent with one another

The data was analyzed by pairwise comparison using the Wilcoxon matched-pairs signed rank test, this is mentioned in the figure legend of the revised manuscript.

We measured IL-1 β and have corrected the mistake in the manuscript text.

8) Figure 8 - PG is also a TLR2 agonist, why was PG alone not sufficient drive inflammatory cytokine expression.

Although peptidoglycan was described as an TLR2 ligand in early studies, subsequent investigations questioned this, see Wolf and Underhill, Nature Rev Immunol 2018, for a comprehensive review on peptidoglycan recognition (<https://www.nature.com/articles/nri.2017.136>). In brief, co-purified cell wall lipoproteins and lipoteichoic acids were likely responsible for the activation effect of peptidoglycan preparations in early studies. Later studies provided evidence that muramyl tripeptides and muramyl tetrapeptides interact with TLR2 but crosslinked (intact) peptidoglycan does not. Furthermore, for *S. aureus* specifically it has been shown that peptidoglycan polymers do not stimulate human TLR2 (<https://pubmed.ncbi.nlm.nih.gov/20522786/>) Thus, the absence of direct stimulation by polymeric *S. aureus* peptidoglycan is not surprising. We have included this information in the results part of the revised manuscript.

Also, they authors need to demonstrate that NOD2 is essential for sensing the PG treated with CwhA, there is no evidence to support the claim that NOD2 is involved.

Unfortunately, we could not obtain NOD2 ko mice within a reasonable time frame (the line is currently not actively bred at Jackson Laboratories, and we found no active breeding colony in our state), and attempts to knock down NOD2 in primary cells were unsuccessful (the methodology is not established in our lab). Thus, we could not experimentally test the hypothesis within a reasonable time frame. While we'll try to follow up on this in the future, this seems out of scope for this manuscript, and, consequently, we have changed the manuscript text to reflect that sensing by NOD2 is speculative and requires further investigations.

Dear Prof. Jacobsen,

Thank you for the submission of your revised manuscript to our editorial offices. I have now received the reports from the three referees that were asked to re-evaluate the study, you will find below. As you will see, the referees now fully support its publication in EMBO reports.

Before we can proceed with formal acceptance, I have these editorial requests I ask you to address in a final revised manuscript:

- Please mark the corresponding author on the title page and provide an e-mail contact.
- Please make sure that all figure panels or tables (main, EV and Appendix figures) are called out separately and sequentially. Presently, there are no separate callouts for panels 8A and 8B. Please check.
- Please change the name of Appendix Figure S1 in its legend (there it is still titled as Supplementary Figure 1).
- We do not allow additional text in the Appendix. Please move the 'Appendix text' (Appendix: General features of peptidoglycan) to the main manuscript text file, where you feel best fit (Introduction, Methods?). Moreover, please move all the related references from the Appendix to the main reference section.
- Please check again that the number "n" for how many independent experiments were performed, their nature (biological versus technical replicates), the bars and error bars (e.g. SEM, SD) and the test used to calculate p-values is indicated in the respective figure legends. Please also check that all the p-values are explained in the legend, and that these fit to those shown in the figure. Please provide statistical testing where applicable. Please avoid the phrase 'independent experiment', but clearly state if these were biological or technical replicates. Please also indicate (e.g. with n.s.) if testing was performed, but the differences are not significant. In case n=2, please show the data as separate datapoints without error bars and statistics. See also:

<http://www.embopress.org/page/journal/14693178/authorguide#statisticalanalysis>

If $n < 5$, please show single datapoints for diagrams. It seems presently many diagrams are missing the 'n.s.'. Moreover:

- Please note that the exact p values are not provided in the legend of figure 5B.
- Please indicate what */ **/ ***/ **** represents; if this represents p value(s), please specify the exact p value in the legend(s) of figure(s) 8A, B.
- Please note that information related to n is missing in the legend of figure 5A
- Please note that the error bars are not defined in the legend of figure 5A.
- Please provide legends for the two movies. Each legend should be provided as a readme.txt file and then each movie should be uploaded together with its legend as a ZIP folder - Movie EV1 and Movie EV2.
- Please add to each legend (main and EV figures, where applicable) a 'Data Information' section (or name the provided section like this) explaining the statistics used or providing information regarding replicates and scales. See:

- The Data Availability Section (DAS) is restricted to information regarding large primary datasets deposited at external databases. Thus, please remove the sentence 'Strains used in this study and the hybridoma cell line used to produce monoclonal antibodies are available upon request.' from the DAS. However, please add information on externally deposited datasets or source data here, provide direct links to the deposited datasets and make sure the datasets are public latest upon online publication of the paper.
- Thank you for providing the requested source data (SD). Please upload this as one folder per main figure (with all files for one figure in one folder and ZIPed together). It seems source data for Fig. 3 and for Fig. 7B is missing. Please check. The uploaded SD for the EV figures is fine (keep it as is). Please add links to the DAS for externally deposited SD.

In addition, I would need from you uploaded separately:

Best,

Referee #1:

All concerns raised have been adequately addressed. No further issues remain.

Referee #2:

The authors have addressed all issues raised by the Reviewers in an adequate and convincing fashion, so it is this reviewer's opinion that this fine manuscript should be accepted and published in EMBO Reports.

Referee #3:

The manuscript is suitable for publication in EMBO reports without further revisions.

All editorial and formatting issues were resolved by the authors.

Prof. Ilse Jacobsen
Leibniz Institute for Natural Product Research and Infection Biology
Research Group Microbial Immunology
Beutenbergstr. 11A
Jena 07745
Germany

Dear Prof. Jacobsen,

I am very pleased to accept your manuscript for publication in the next available issue of EMBO reports. Thank you for your contribution to our journal.

Yours sincerely,
